# Assessing the causal association of glycine with risk of cardio-metabolic diseases

Laura B.L. Wittemans[1], Luca A. Lotta[1], Clare Oliver-Williams[2,3], Isobel D. Stewart[1], Praveen Surendran[2], Savita Karthikeyan[2], Felix R. Day [1], Albert Koulman [1,4], Fumiaki Imamura [1], Lingyao Zeng[5,6], Jeanette Erdmann[7,8,9], Heribert Schunkert[5,6], Kay-Tee Khaw[10], Julian L. Griffin[11], Nita G. Forouhi [1], Robert A. Scott[1], Angela M. Wood[2], Stephen Burgess [2,12], Joanna M.M. Howson [2], John Danesh[2,13], Nicholas J. Wareham[1], Adam S. Butterworth [2] & Claudia Langenberg [1]

Circulating levels of glycine have previously been associated with lower incidence of coronary heart disease (CHD) and type 2 diabetes (T2D) but it remains uncertain if glycine plays an aetiological role. We present a meta-analysis of genome-wide association studies for glycine in 80,003 participants and investigate the causality and potential mechanisms of the association between glycine and cardio-metabolic diseases using genetic approaches. We identify 27 genetic loci, of which 22 have not previously been reported for glycine. We show that glycine is genetically associated with lower CHD risk and find that this may be partly driven by blood pressure. Evidence for a genetic association of glycine with T2D is weaker, but we find a strong inverse genetic effect of hyperinsulinaemia on glycine. Our findings strengthen evidence for a protective effect of glycine on CHD and show that the glycine-T2D association may be driven by a glycine-lowering effect of insulin resistance.

[1] MRC Epidemiology Unit, University of Cambridge, Cambridge CB2 0QQ, UK. [2] MRC/BHF Cardiovascular Epidemiology Unit, Department of Public Health and Primary Care, University of Cambridge, Cambridge CB1 8RN, UK. [3] Homerton College, Hills Road, Cambridge CB2 8PH, UK. [4] NIHR BRC Nutritional Biomarker Laboratory, University of Cambridge, Cambridge CB2 0QQ, UK. [5] Deutsches Herzzentrum München, Technische Universität München, München 80636, Germany. [6] DZHK, Partner Site München Heart Alliance, München 80802, Germany. [7] Institute for Cardiogenetics, University of Lübeck, Lübeck 23562, Germany. [8] DZHK, partner site Hamburg/Lübeck/Kiel, Lübeck 23562, Germany. [9] University Heart Center Lübeck, Lübeck 23562, Germany. [10] Department of Public Health and Primary Care, Institute of Public Health, University of Cambridge, Cambridge CB2 0SR, UK. [11] Biochemistry Department, University of Cambridge, Cambridge CB2 1QW, UK. [12] MRC Biostatistics Unit, University of Cambridge, Cambridge CB2 0SR, UK. [13] Wellcome Sanger Institute, Genome Campus, Hinxton CB10 1SA, UK. Correspondence and requests for materials should be addressed to C.L. (email: Claudia.Langenberg@mrc-epid.cam.ac.uk)

C irculating levels of glycine, a non-essential amino acid involved in a wide range of metabolic pathways[1], have been associated with lower incidence of myocardial infarction (MI)[2] and type 2 diabetes (T2D)[3–6] in large-scale epidemiological studies. A genetic study suggested a sex-specific role of glycine metabolism on risk of coronary heart disease (CHD)[7], based on a genetic variant in carbamoyl-phosphate synthase 1 (CPS1) which was associated with higher risk of CHD in women but not in men. However, this study did not take into account the stronger effect size of CPS1 on glycine in women[8]. A recent Mendelian randomisation (MR) study suggested that low glycine levels may be causally related to T2D risk, based on a genetic score which was largely driven by CPS1[3]. Due to pleiotropic effects of this locus, this association may have been driven by glycine-independent mechanisms. Uncertainty about the role of glycine metabolism on cardio-metabolic disease risk and its potential sex-specific nature therefore remains.

Genome-wide association studies (GWAS) for plasma glycine levels aiming to identify genetic regions involved in glycine metabolism have been previously conducted in up to 25,000 participants, identifying five genetic loci associated with glycine levels[9–14]. Potential sex differences in the effect sizes of these loci have not been systematically assessed. A larger GWAS on glycine could increase the number of genetic loci robustly associated with glycine, thereby improving our understanding of the genetic determinants of glycine metabolism and providing additional genetic instruments to assess the causality of glycine on cardio-metabolic diseases and related risk factors, using the MR framework[15].

We here present a genetic discovery for glycine and identify 22 genetic loci which are novel for glycine. We construct and validate sex-combined and sex-specific genetic scores for glycine with different degrees of specificity to the glycine pathway. We find that genetically predicted glycine is significantly associated with CHD risk in men and women, which is in line with a causal role for glycine and CHD risk, and identify blood pressure as a potential mediator. We furthermore identify significant effects of specific genetic variants in genes related to glycine catabolism on T2D but evidence for an overall causal role of glycine levels on T2D is weak. Genetically predicted insulin resistance (IR) is strongly associated with lower glycine, which may drive the consistently observed association between higher glycine levels and lower incidence of T2D.

## Results

**22 novel genetic loci for glycine.** We conducted a $Z$ score-based meta-analysis of GWAS for glycine levels in up to 80,003 participants of European ancestry, including 55,673 participants from the Fenland, INTERVAL and EPIC-Norfolk studies, and two publicly available summary-level GWAS datasets (Supplementary Table 1). 27 genetic loci were identified for glycine ($p < 5 \times 10^{-8}$), of which 22 have not previously been reported for glycine (Fig. 1, Supplementary Data 1). A total of 20 secondary signals at 8 loci were identified through approximate conditional association analyses (Supplementary Data 2). Six of the 27 loci are in (or near) genes encoding enzymes involved in glycine metabolism (Fig. 2). No evidence for heterogeneity between the studies was found, except for the strongest locus, CPS1 (Supplementary Data 1, Supplementary Figure 1).

Thirteen of the 27 glycine loci have been reported for cardio-metabolic traits and risk factors, including blood lipid fractions[16], blood coagulation[17–22] and glycaemic traits[23], T2D[24] and CHD[25] (Supplementary Data 3). However, the direction of association with the cardio-metabolic traits relative to the glycine-raising allele was not consistent between loci, suggesting that these pleiotropic loci do not reflect an overall shared genetic control of glycine and cardio-metabolic traits.

Rs715 near CPS1 had by far the strongest effect on glycine (per-allele beta ± standard error (SE) on standard deviations (SDs) of glycine = 0.444 ± 0.006, effect allele frequency (EAF) = 31.3%) and explained 13.7% of the variance. This variant is in high linkage disequilibrium (LD) ($r^2 = 0.904$) with a missense variant (rs1047891). Conditioning on rs1047891 in EPIC-Norfolk showed that the effect of rs715 was entirely driven by rs1047891 (beta ± SE for rs715 on glycine before adjustment = 0.565 ± 0.013; after adjustment for rs1047891: beta = 0.032 ± 0.049).

Previous studies highlighted sex differences in the effect size of the CPS1 locus on glycine[8]; we therefore assessed sex-specific effect sizes of all 27 loci on glycine levels standardised by sex in the Fenland, EPIC-Norfolk and INTERVAL studies (Bonferroni-corrected $p$ for sex difference based on 27 tests and two-sided $t$-test: $p_{sex} < 0.002$). We confirmed that the effect of the CPS1 locus on glycine is nearly three-fold stronger in women than in men (women: beta ± SE = 0.691 ± 0.009, men: beta = 0.233 ± 0.007, $p$ for sex difference based on two-tailed $t$-test: $p_{sex} < 2 \times 10^{-302}$) and found a similar sex difference in effect sizes on 65 other metabolites significantly associated with rs715 in EPIC-Norfolk (Supplementary Figure 2). Significant sex differences were also found for rs17591030 (GLDC, $p_{sex} = 3.3 \times 10^{-7}$) and rs10184004 (TRIB1, $p_{sex} = 0.001$) (Fig. 3a, Supplementary Data 1). The cumulative variance in glycine explained by the 27 variants was nearly 2.5 times higher in women (25.1%) than in men (10.6%), which was mostly driven by the larger effect size of rs715 in women (Fig. 3b).

**Genetic scores for glycine: power versus specificity.** The effect size-weighted genetic score including all 24 common loci for glycine explained 15.6% of the total variance in both sexes combined, and 10.0% and 24.9% in men and women, respectively. As a comparison, sex and BMI explained 8.6% and 3.4% of the total variance in glycine levels, respectively, while other modifiable risk factors explained a very small proportion (alcohol consumption: 0.51%, smoking: 0.08%) or none of the variance (age and physical activity). The low-frequency (TUBGCP4) and rare variants (FGG and KIF5B) were excluded from the genetic score as look-ups for these were not available in all summary-level GWAS datasets.

All 24 common loci were within 1 Mb of or in LD with loci previously reported for other metabolites (Supplementary Data 3). We therefore tested to what extent the glycine score affected metabolic pathways unrelated to glycine metabolism. Of the 894 metabolites available in EPIC-Norfolk, the 24 SNP score was most strongly associated with glycine but also affected 73 other metabolites (Bonferroni-corrected $p$ for 894 tests based on two-sided $t$-test: $p < 5.6 \times 10^{-5}$), of which the majority were strongly correlated with and/or metabolically related to glycine (e.g., urea cycle and choline metabolites, serine, glycine-conjugated fatty acids). The 24 SNP score was also associated with several metabolites with no known link to glycine, including phospho-lipids and unknown metabolites (Supplementary Data 4).

We constructed three additional scores for glycine with decreasing numbers of loci but increasing specificity to glycine metabolism. The 6 SNP score was comprised of the six loci near genes encoding enzymes related to glycine metabolism (Fig. 2), explained 14.9% of the variance and remained associated with 70 metabolites (Supplementary Data 4). The CPS1 locus by itself was significantly associated with all but seven of these 70 metabolites (Supplementary Data 4). We therefore generated a third score that excluded the pleiotropic CPS1 locus. This score explained 1.2% of the glycine variance and was associated with 13 metabolites, of which eight had a clear metabolic link to glycine

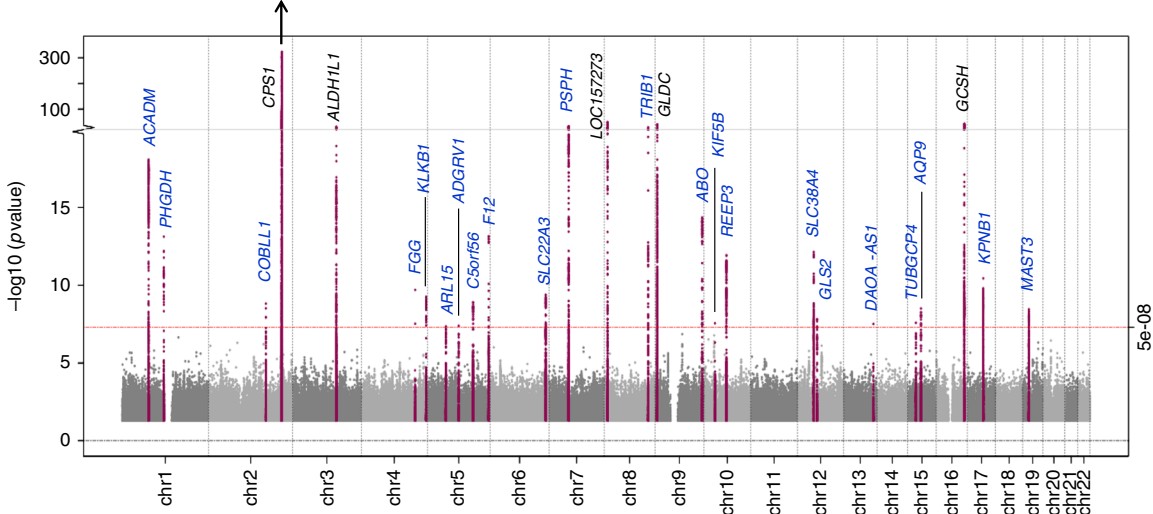

**Fig. 1** Manhattan plot of *Z* score-based meta-analysis of GWAS for glycine levels. Loci in blue have not previously been reported for glycine. *P*-value for *CPS1* locus = $3 \times 10^{-1632}$

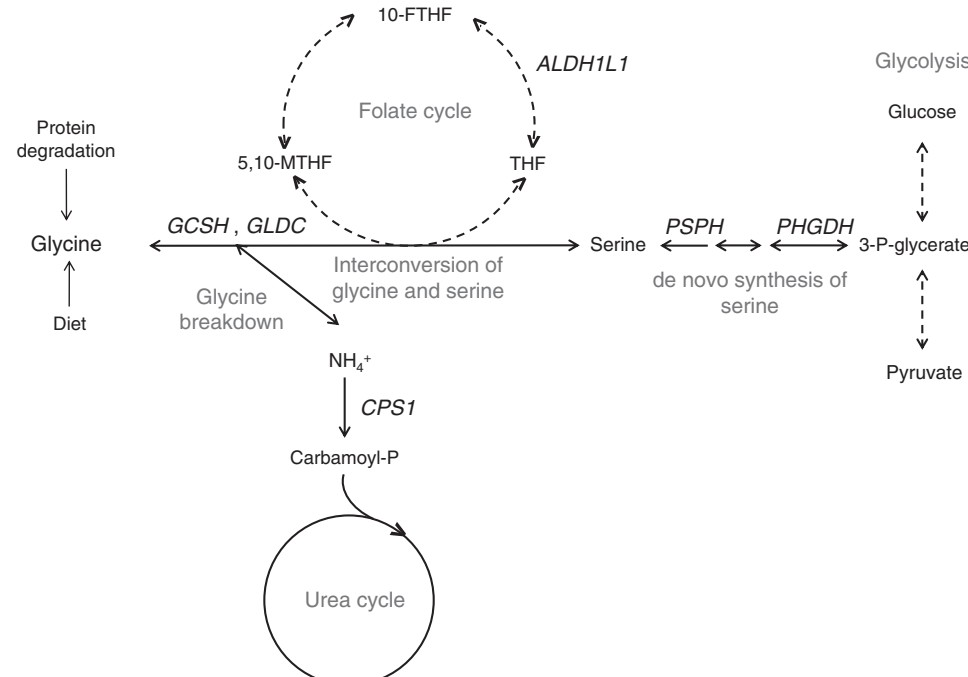

**Fig. 2** Schematic overview of glycine metabolism and six genetic loci for glycine in/near genes encoding enzymes related to glycine metabolism. *GLDC* (glycine decarboxylase) and *GCSH* (glycine cleavage system protein H) are part of the glycine cleavage system, the major enzyme complex responsible for glycine degradation. *PHGDH* (phosphoglycerate dehydrogenase) and *PSPH* (phosphoserine phosphatase) encode enzymes involved in the de novo biosynthesis of serine, which can be converted into and synthesised from glycine. The interconversion of glycine and serine serves as an important source of methyl groups in the folate cycle, a metabolic pathway of which the enzyme encoded by *ALDH1L1* (aldehyde dehydrogenase 1 family member L1) is a component. *CPS1* (carbamoyl-phosphate synthase 1) encodes the enzyme responsible for the rate-limiting step of the urea cycle which is responsible for the detoxification of ammonia. Glycine breakdown produces ammonia; changes in the efficiency of the urea cycle are therefore likely to have upstream effects on glycine metabolism. THF: tetrahydrofolate, 5,10-MTHF: $N^5$-$N^{10}$-methylenetetrahydrofolate, 10-FTHF: 10-formyltetrahydrofolate

(γ-glutamylglycine, N-acetylglycine, propionylglycine, N-palmitoylglycine, isovalerylglycine, serine and cinnamoylglycine) and four were unknown metabolites. Finally, we constructed a 2 SNP score, based on the loci at *GCSH* and *GLDC* encoding enzymes of the glycine cleavage system, the major enzyme complex responsible for glycine breakdown. The 2 SNP score explained 0.6% of the variance in glycine and was associated with γ-glutamylglycine, N-acetylglycine, propionylglycine, isovalerylglycine, 3-methylglutaconate and its carnitine conjugate, and one unknown metabolite (Supplementary Data 4).

**Glycine is genetically associated with lower CHD risk**. Based on MR analyses in up to 88,800 CHD cases and 485,266 controls and using the 24 SNP score, we estimated that each SD higher genetically predicted glycine was associated with an odds ratio (OR) of 0.95 for CHD. ([95% confidence intervals (CI)] = [0.92, 0.98], *p* based on two-sided *t*-test: $p = 0.001$) (Fig. 4, Supplementary Table 2). Sex-specific analyses in up to 9852 female and 21,994 male CHD cases showed that the effect sizes of glycine on CHD risk were similar in women and men (*p* for sex difference based on two-sided *t*-test: $p_{\text{sex}} = 0.60$) (Fig. 4 and Supplementary

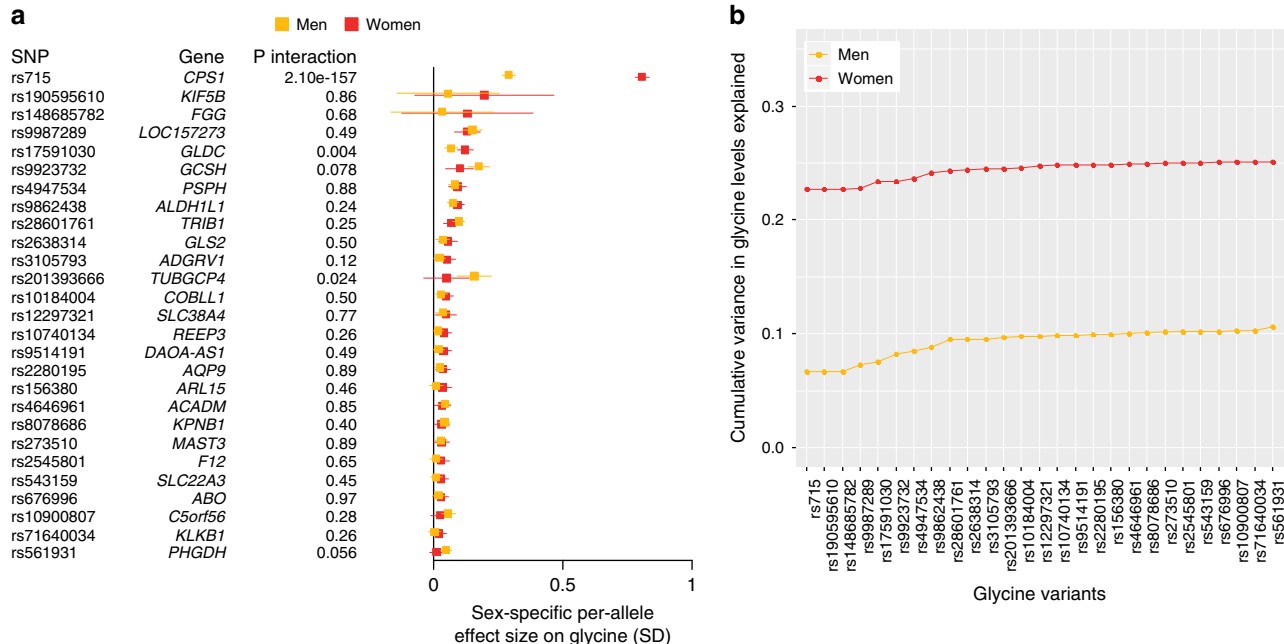

**Fig. 3** Sex-specific effect sizes and cumulative variance explained by lead SNPs at the 27 genetic loci for glycine. **a** Sex-specific effect sizes of lead SNPs at the significant loci for glycine levels based on a meta-analysis of the Fenland, EPIC-Norfolk and INTERVAL studies, including 30,226 men and 31,957 women. **b** Cumulative variance in glycine levels explained ($r^2$) by lead SNPs at all 27 loci in men ($N = 5086$) and women ($N = 5706$) of the EPIC-Norfolk study (sub-cohorts A and B)

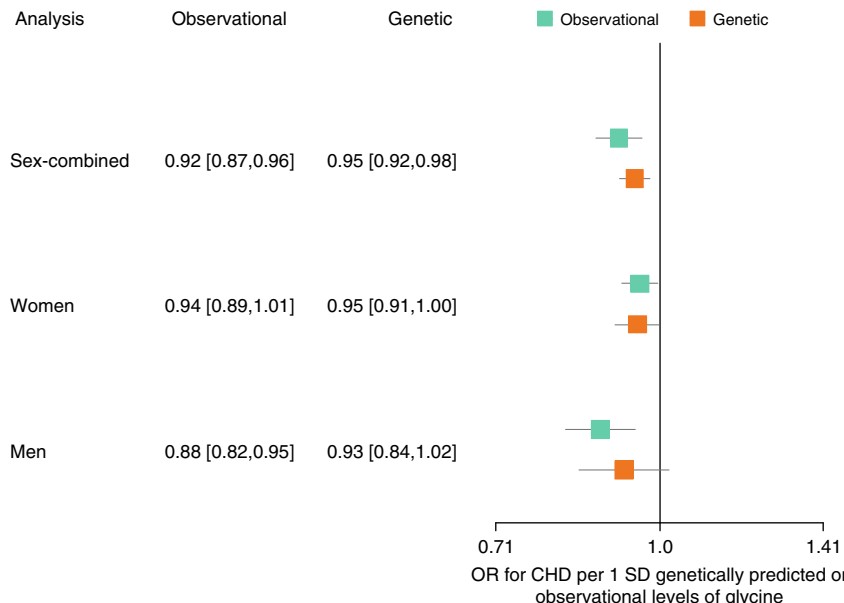

**Fig. 4** Forest plot of the odds ratios ± 95% confidence intervals for coronary heart disease per standard deviation observationally or genetically predicted higher levels of glycine. Genetic estimates are based on 88,800 CHD cases and 485,266 controls. Sex-specific genetic analyses included 9,852 female and 21,994 male CHD cases and 202,124 female and 164,944 male controls. For the sex-specific analyses, sex-specific standard deviations for glycine were used, which were 0.321 and 0.195 (arbitrary units) for women and men in EPIC-Norfolk, respectively. Observational analyses are based on 11,147 participants (4989 men and 6158 women), of which 2053 (1223 men and 830 women) were incident CHD cases

Figures 3 and 4). After taking into account the sex-specific effect on glycine, the genetic glycine-CHD association based on *CPS1* only did also not differ by sex (women: OR [95% CI] = 0.96 [0.91,1.00], men: OR [95% CI] = 0.94 [0.85,1.03], $p_{\text{sex}} = 0.74$).

Visual inspection of the scatter plots (Fig. 5), statistical tests for heterogeneity and pleiotropy (Egger's intercept $p = 0.003$; men-only: Cochran's Q $p = 0.09$) (Supplementary Table 2) and the presence of several variants within the score associated with other

traits, suggest that the effect sizes of the 24 glycine-raising genetic variants on CHD risk may be heterogeneous. To reduce the influence of potential pleiotropic variants on the causal estimate, we used the weighted median MR method for the main analyses. To verify if the inverse association of genetically predicted glycine with genetic risk of CHD was driven by glycine-unrelated pathways, we conducted a series of sensitivity analyses using the three genetic scores that are subsets of the 24 SNP score and have

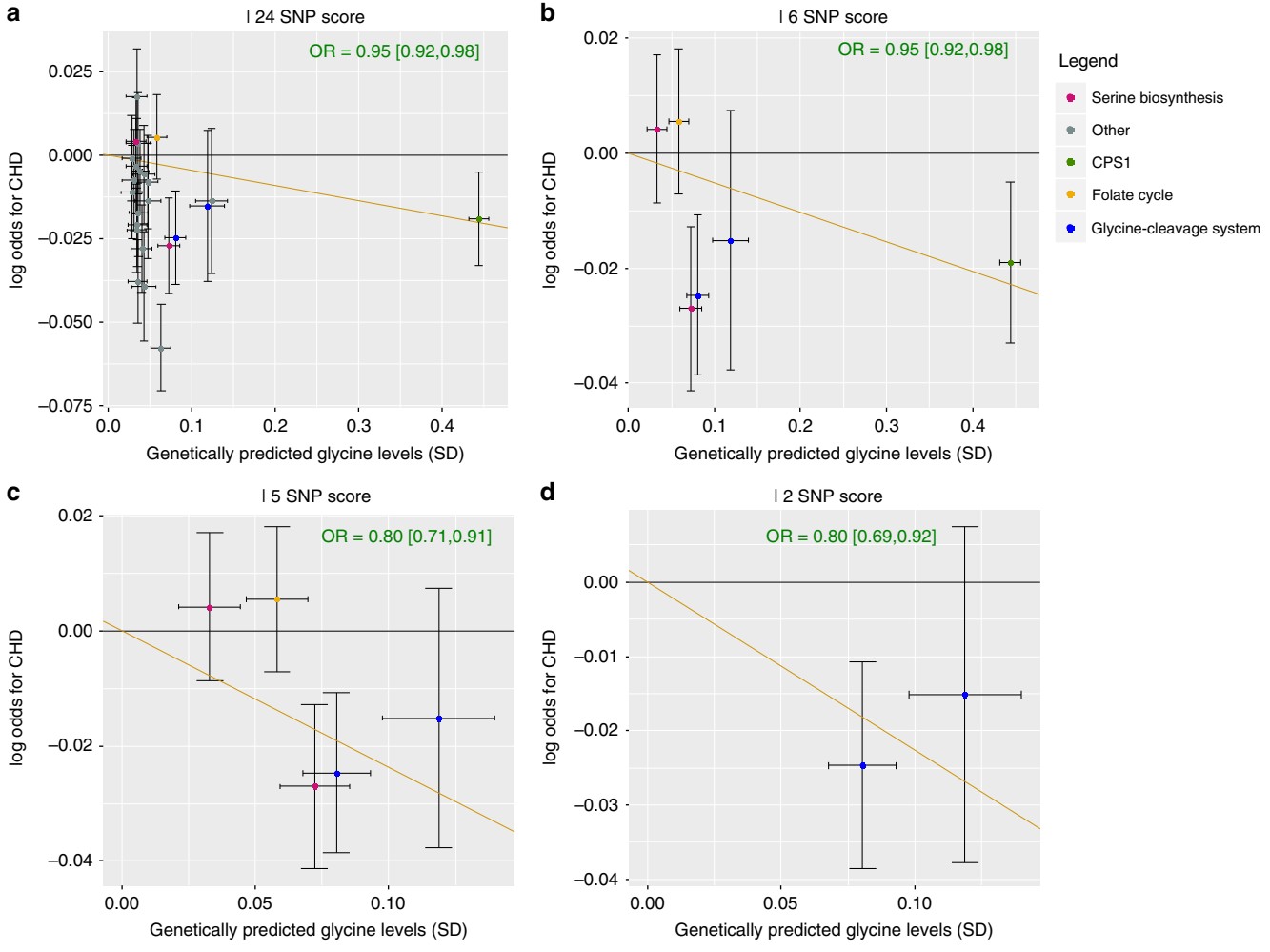

**Fig. 5** Scatter plots of the effect sizes ± 95% confidence intervals of genetic variants for glycine on standard deviations of glycine levels versus the log odds for coronary heart disease. **a** For the 24 SNP score, **b** 6 SNP score, **c** 5 SNP score and **d** the 2 SNP score. The orange line represents the slope estimated using the weighted median method

increasing specificity to glycine metabolism. When using the 6 SNP score which included loci with a known biological link to glycine, similar effect estimates for glycine on CHD risk were found and the heterogeneity decreased (Egger's intercept $p = 0.44$; men-only: Cochran's Q $p = 0.80$) (Fig. 5b). Removing the strong but pleiotropic *CPS1* locus from the 6 SNP score increased the causal effect estimate of glycine on CHD risk (sex-combined: OR = 0.80 [0.71,0.91], $p = 4.8 \times 10^{-4}$) (Fig. 5c), and a similar effect estimate was obtained when using the 2 SNP score (*GCSH* and *GLDC*) (sex-combined: OR = 0.80 [0.69,0.92], $p = 1.7 \times 10^{-3}$) (Fig. 5d, Supplementary Table 2, Supplementary Figures 3 and 4). No significant associations of genetically predicted glycine with stroke or with stroke sub-types were found (Supplementary Figure 5, Supplementary Table 3).

In a non-genetic analysis using Cox proportional hazards models adjusted for sex, glycine levels were associated with lower risk of CHD and MI (CHD: hazard ratio (HR) for CHD per 1SD higher glycine = 0.92, $p$ based on two-sided $t$-test: $p = 4.7 \times 10^{-4}$; MI: HR = 0.90, $p = 0.005$). This observational association was similar to the genetically predicted association based on the 24 SNP score for glycine (Fig. 4). After adjustment for cardiometabolic risk factors, the associations of glycine were only modestly (CHD) or not (MI) attenuated (CHD: HR = 0.95, $p = 0.058$; MI: HR = 0.90, $p = 0.024$). However, no association with stroke was found (Supplementary Table 4).

**Biological pathways mediating the effect of glycine on CHD**. To explore the biological mechanisms through which glycine may influence risk of CHD, we investigated the downstream effects of genetic differences in glycine levels on CHD risk factors, including systolic (SBP) and diastolic blood pressure (DBP), blood lipid fractions (triglycerides, HDL, LDL and total cholesterol) and 13 potentially relevant blood cell traits (Bonferroni-corrected $p$ for 19 tests based on two-sided $t$-test: $p < 2.6 \times 10^{-3}$). Genetically predicted glycine based on the 24 SNP score was significantly associated with lower genetically predicted SBP (beta ± SE per SD glycine on SD of SBP = $-0.028 \pm 0.007$, $p = 1.5 \times 10^{-5}$) and nominally with lower genetically predicted DBP (beta = $-0.019 \pm 0.009$, $p = 0.039$). Similar effect sizes were found for men and women separately (Fig. 6, Supplementary Table 5, see Supplementary Figure 6 for sex-specific associations for rs715). Effect estimates using the 6 SNP score were similar as when using the 24 SNP score. When using the 5 and 2 SNP scores for glycine, effect sizes tended to increase (Fig. 6, Supplementary Table 5).

Glycine was not genetically associated with blood lipids and blood cell traits. Associations with lower HDL cholesterol and five blood cell traits reached significance when using the 24 SNP score, but the effect sizes drastically decreased when using the 5 or 2 SNP scores, suggesting that the associations were driven by *CPS1* only (Supplementary Table 5, Supplementary Figure 7).

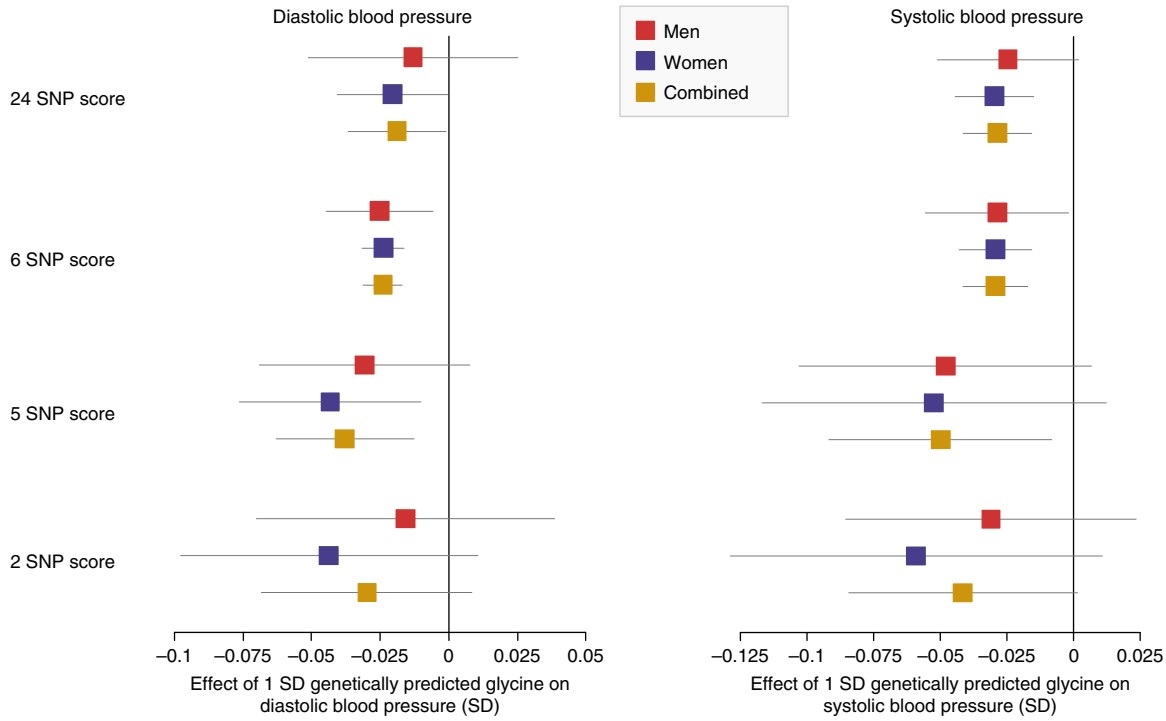

**Fig. 6** Genetically predicted effect size ± 95% confidence intervals of glycine on diastolic and systolic blood pressure, in sex-combined and sex-specific analyses, using four different genetic scores for glycine. Associations are based on 203,943 male and 241,417 female European ancestry UK biobank participants

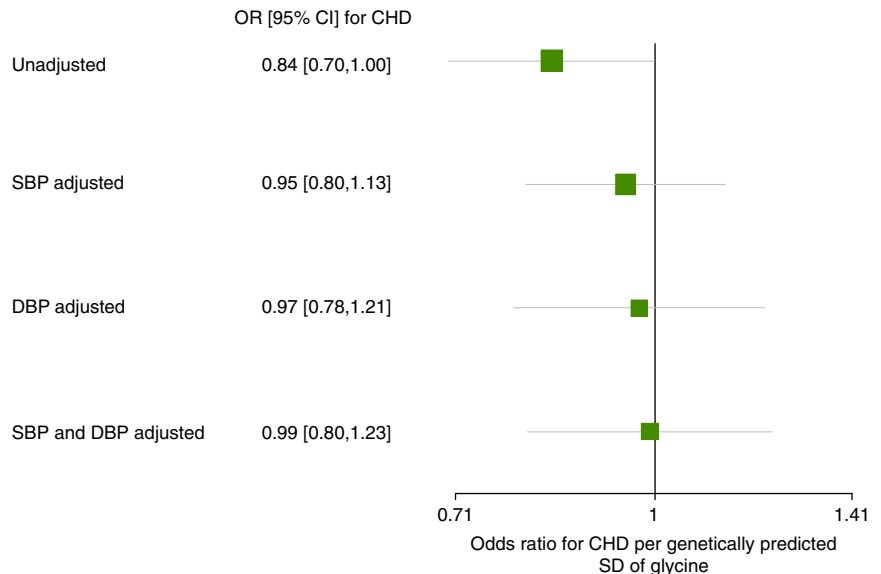

**Fig. 7** Forest plot of the odds ratios ± 95% confidence intervals for coronary heart disease per genetically predicted standard deviation of glycine, with and without adjustment for blood pressure. CHD: coronary heart disease, DBP: diastolic blood pressure, SBP: systolic blood pressure, SD: standard deviation, CI: confidence interval

To test if the genetic association of glycine with CHD was mediated through lowering blood pressure, we adjusted the effect of genetically predicted glycine on CHD for genetically predicted SBP and DBP. We used the 5 SNP glycine score to reduce the likelihood of glycine-unrelated mechanisms. Adjusting for SBP and DBP separately and together progressively reduced the effect estimate of genetically predicted glycine on CHD risk (Fig. 7). This suggests that the genetic association of glycine with CHD risk may be mediated through blood pressure.

**The association of glycine with T2D may be pathway-specific**. Based on up to 74,124 T2D cases and 824,006 controls and the 24 SNP score, genetically predicted glycine was not associated with T2D risk (OR [95% CI] for T2D per genetically predicted SD higher glycine = 0.99 [0.96, 1.02], $p$ for two-sided $t$-test: $p = 0.59$) (Fig. 8, Supplementary Table 6). Sex-specific analyses in up to 12,013 female and 16,914 male T2D cases indicated that glycine was also not genetically associated with T2D in men or women only. Similar effect estimates were found across all 4 MR methods and when using the 6 SNP score (Fig. 8, Supplementary Table 6).

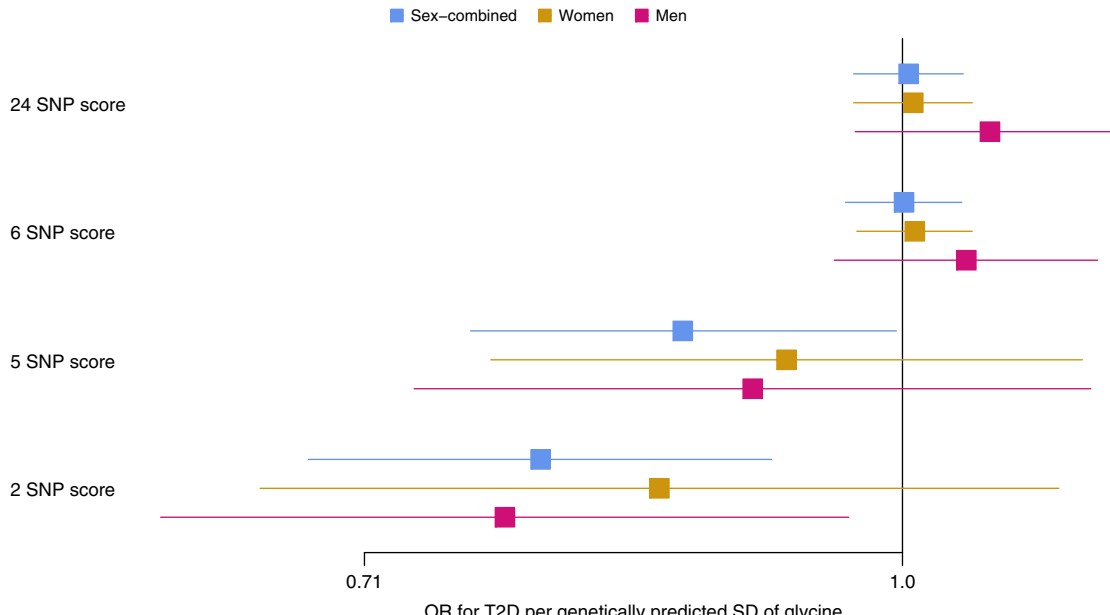

**Fig. 8** Forest plot of the odds ratios ± 95% confidence intervals for type 2 diabetes per standard deviation genetically predicted higher levels of glycine using four different genetic scores for glycine. Genetic estimates are based on 74,124 T2D cases and 824,006 controls for sex-combined analyses. Sex-specific genetic analyses include 12,013 female and 16,914 male T2D cases and 202,124 female and 164,944 male controls. For the sex-specific analyses, the standard deviations of the sex-specific glycine distributions were used, which were 0.321 and 0.195 (arbitrary units) for women and men in the EPIC-Norfolk study, respectively. Results based on the weighted median MR method are shown

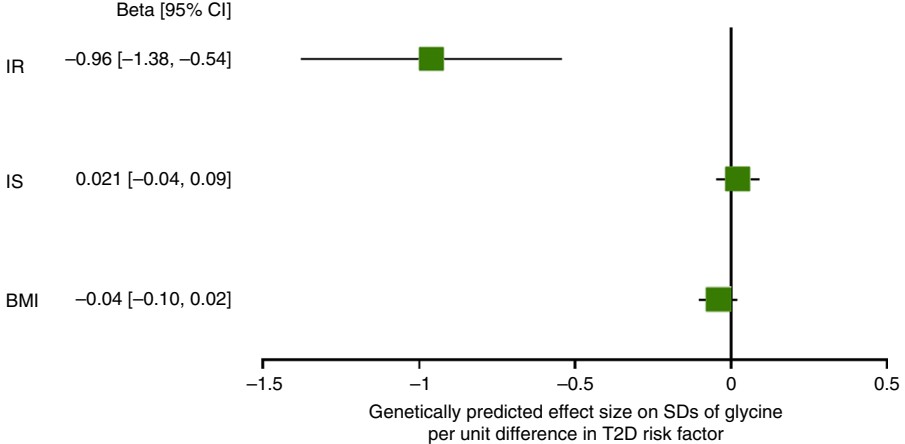

**Fig. 9** Forest plot of the genetic effect sizes ± 95% confidence intervals of insulin resistance (IR), early-phase insulin secretion (IS) and body mass index (BMI) with glycine levels. Analyses were based on the inverse variance-weighted MR method

However, when using the 5 and 2 SNP scores which excluded *CPS1*, the association of glycine with T2D reached significance (Sex-combined OR based on 5 SNP score = 0.85 [0.76,0.96], $p$ = 0.007; OR based on 2 SNP score = 0.82 [0.72,0.94], $p$ = 0.004) (Supplementary Figures 8-10, Supplementary Table 6). No evidence for heterogeneity by sex was found ($p$ for sex difference based on two-sided $t$-test: 5 SNP score: $p_{sex}$ = 0.88; 2 SNP score: $p_{sex}$ = 0.56). In an observational analysis, we replicated the strong observational association of glycine levels with lower incidence of T2D (HR [95% CI] for T2D per SD of glycine = 0.58 [0.47, 0.70]).

**Hyperinsulinaemia as a driver of low glycine levels**. To investigate if the association between low levels of glycine and high incidence of T2D could be a consequence of early disease processes of T2D, we assessed the genetically predicted effects of three T2D risk factors—elevated body mass index (BMI), reduced early-phase insulin secretion (IS) and IR—on levels of glycine. We found that genetically higher IR was strongly associated with genetically lower levels of glycine (beta [95%CI] on SDs of glycine levels per genetically predicted unit of fasting insulin levels = −0.96 [−1.38, −0.54], $p$ based on two-sided $t$-test: $p = 5.98 \times 10^{-6}$), while genetically predicted BMI (beta per SD genetically predicted BMI on SDs of glycine levels = −0.04 [−0.10,0.02], $p$ = 0.169) and IS (beta per genetically predicted units of insulin at 30 min during an oral glucose tolerance test on SDs of glycine levels = 0.02 [−0.04, 0.09], $p$ = 0.52) were not associated with glycine (Fig. 9). No evidence for heterogeneity or directional pleiotropy was found for any of the three traits. The inverse-variance weighted MR analyses were therefore used as the main analyses, but similar effect sizes were found using other MR methods (Supplementary Table 7).

**Genetically higher glycine not linked to higher cancer risk**. To test whether higher levels of glycine may increase cancer risk, we used publicly available summary statistics from the BCAC[26],

OCAC[27] and PRACTICAL[28] GWAS consortia to assess associations of genetically predicted glycine with risk of breast, ovarian and prostate cancer. We found no evidence for an increased risk for these three site-specific cancers for any of the four genetic scores for glycine (Supplementary Table 8).

## Discussion

Based on a genetic discovery in up to 80,003 participants from five studies, we identify 22 loci not previously reported for glycine. Two novel loci (PSPH and PHGDH) are located in genes encoding enzymes catalysing the de novo biosynthesis of serine—a pathway which has not previously been genetically associated with glycine, despite the close metabolic link between glycine and serine. We also identified two rare variants (in FGG and near KIF5B) not previously reported for metabolic traits, which corroborates previous evidence that rare genetic variation plays a role in amino acid levels[13].

Based on four genetic scores with different degrees of specificity to the glycine pathway, we demonstrate that low levels of glycine are genetically associated with higher risk of CHD, which supports a potential protective role of glycine against CHD. The effect size differs depending on the genetic score, with estimated effects of glycine on CHD increasing with the degree of specificity of the genetic score to glycine, which strengthens the evidence that the observed genetic associations are driven by glycine. We furthermore found that glycine has similarly protective effects in men and in women, which contradicts findings by Hartiala et al.,[7] who suggested that the sex-specific association of CPS1 with CHD may reflect a causal role for glycine on CHD in women only. We however show that, after taking into account the sex-specific effect sizes on glycine, the genetic association of glycine with CHD, based on CPS1 only and the four genetic scores for glycine, is similar in magnitude in both sexes.

Our findings suggest that the inverse genetic association of glycine with blood pressure may at least partly explain the genetic glycine-CHD association. High blood pressure is a major risk factor for cardiovascular diseases[29] and glycine supplementation has been shown to lower blood pressure in rodents[30–32] and in one small intervention study on human participants[33]. Our findings corroborate a potential blood pressure-lowering effect of glycine for the first time in a large-scale epidemiological setting. A proposed mechanism through which glycine may lower blood pressure is by the glycine-gated chloride channels expressed on the endothelial surface, which upon binding of glycine induce membrane hyperpolarisation and thus the production of nitric oxide—a well-known vasodilator[34]. Despite the strong genetic association of glycine with blood pressure, we cannot conclude that the genetic effect of glycine on CHD risk is entirely mediated through blood pressure, due to the limited power of the multivariable MR analyses. Moreover, the absence of associations between glycine and stroke, for which hypertension is also a major risk factor[29], suggests that there also may be other physiological pathways through which glycine influences cardiovascular disease risk. We however find no evidence for a role of glycine in lipid metabolism or blood cell traits, despite previous studies suggesting that glycine may regulate lipid metabolism[2] and platelet[35] and immune cell activation[36]. There may be other mechanisms underlying the genetic glycine-CHD association which we could not investigate here. For example, glycine may protect against oxidative stress, as it is a substrate for the biosynthesis of glutathione—a major antioxidant in human cells[37,38].

While multiple observational epidemiological studies have linked high glycine levels to lower incidence of T2D[3,4,39], the genetic evidence for causality is weaker. MR analyses based on the

larger but less specific genetic scores indicate no causal effect of glycine on risk of T2D, while variants in genes related to glycine catabolism drive associations of the 5 and 2 SNP scores with lower T2D risk. These findings suggest that overall levels of glycine may not be causal for T2D but that low glycine levels due to changes in the catalytic efficiency of the glycine cleavage system—the protein complex that catalyses the major catabolic pathway of glycine[40]—may cause higher risk of T2D. Coding mutations in genes of the glycine cleavage system have been identified as one of the causes of severe hyperglycinemia[41], a life-threatening inborn error of metabolism, and have been linked to higher risk of neural tube defects[42]. To the best of our knowledge, this enzyme complex has not previously been linked to common cardio-metabolic diseases. An alternative interpretation of our findings is that overall glycine levels are causally associated with T2D risk, but that this association is masked by pleiotropic effects of CPS1. As CPS1 had a 10-fold stronger effect on glycine than most other glycine-associated variants, the genetic scores which included CPS1 in fact mostly represented CPS1. The CPS1 locus has widespread effects across the metabolome, which are not restricted to the glycine pathway, and has also been associated with other traits, of which some, including glycine, may protect against T2D while others could increase T2D risk. Therefore, the null-association based on the genetic scores including CPS1 may have been driven by glycine-independent pathways and may thus mask a true genetic association of glycine with T2D.

Our findings do not entirely replicate the recently reported protective effect of glycine on T2D based on an MR analysis using five loci for glycine[3]. The significant inverse genetic association of glycine with T2D risk reported by Merino et al. was largely driven by CPS1, of which the glycine-raising allele was nominally associated with lower T2D risk in the 11,600 T2D cases and 33,000 controls. Our analyses based on 74,124 T2D cases and 824,006 controls did not replicate the nominal association with T2D for CPS1, nor for the genetic scores that included CPS1.

We find strong genetic evidence that low glycine levels are a consequence of IR, which suggests that the inverse association of glycine levels with incidence of T2D is at least partly driven by IR. Observational epidemiological research has previously described that measures of IR are inversely associated with glycine[43,44] and rodent experiments suggest that increased oxidative stress caused by IR leads to a higher rate of glutathione synthesis, which may lead to glycine depletion[45]. Another proposed mechanism that could link IR to lower levels of glycine is the increased demand for glycine to safely excrete fatty acyl-CoA esters which accumulate as a consequence of incomplete suppression of lipolysis and resulting high circulating levels of fatty acids in the insulin resistant state[46]. Free fatty acids can lead to a build-up of beta oxidation intermediates, i.e., fatty acyl-CoA esters, which can upon trans-conjugation with glycine be safely excreted as fatty acyl-glycine[46,47].

Our findings raise the question whether glycine supplementation merits evaluation as a strategy to prevent cardio-metabolic disease. A variety of glycine supplements are available for purchase "over the counter" and are used without clinical follow-up for a wide range of indications. To consider the preventive value, the potential benefits and harms should be assessed. Effect sizes for CHD varied depending on the "specificity" of the genetic score and, in addition, CI across these estimates varied, ranging from 0.69 to 0.98 per SD higher glycine levels. However, the increase in the effect estimate of glycine with increasing specificity of the genetic score suggests that the benefit may be considerable. This, together with the feasibility of achieving the required glycine concentration in blood given its short half-life[48], and any potential harms arising, require piloting under controlled conditions. In terms of safety, in vitro and animal-based evidence has

suggested that glycine and serine may promote oncogenesis by fuelling one-carbon and folate metabolism[49]. This is indirectly supported by a suggestive association between folic acid supplementation and cancer incidence[50]. Furthermore, it was recently shown that a serine and glycine-depleted diet improves treatment response and survival in murine models of colorectal cancer and lymphoma[51]. Our genetic analyses of three site-specific cancers where GWAS summary statistics were available to provide maximum power and specificity did not provide evidence that glycine levels increase the risk of breast, ovarian or prostate cancer. However, consortia data for other cancer sites and types are required to provide genetic evidence against an increased risk for site-specific cancers other than breast, ovarian and prostate, and a thorough assessment of the potential carcinogenicity of glycine is required before glycine supplementation can be considered even in an evaluative setting in human participants.

Our investigation into the aetiological role of glycine metabolism in cardio-metabolic diseases has several strengths. The variance in glycine levels explained by the genetic score comprising 24 loci was more than 15%, which enabled us to conduct well-powered MR experiments. The high proportion of explained variance can be attributed to the biological specificity of the exposure. Secondly, we conducted a thorough assessment of pleiotropic effects of the genetic scores for glycine and generated a series of subsets of the full genetic score with increasing specificity to glycine metabolism, in order to reduce the likelihood of contributions of glycine-unrelated metabolic pathways to the causal effect estimate. We adopted multiple MR approaches, including robust methods which are less sensitive to bias due to pleiotropic effects by a subset of genetic variants within the score[52–54], which further decreases the chance that the genetically determined effect estimate was driven by glycine-independent pathways.

We are aware of certain limitations to our study. First, due to differences in imputation reference panels between the studies included in the GWAS for glycine, sample sizes differed between the tested variants. Therefore certain loci, in particular those only covered in UK10K and rare variants, which have been shown to influence amino acid levels[13], may not have reached significance. Secondly, some of the studies upon which our analyses were based were enriched for healthy participants, e.g., UK Biobank[55] and INTERVAL. It has recently been suggested that selection bias could theoretically lead to a false positive genetic association between an exposure and an outcome if both the exposure and the outcome influence the likelihood of an individual participating in the study[56]. As individuals are not aware of their glycine levels, any such selection would have to be indirect through other factors that affect glycine and bias participation in each of the independent studies included in the same direction. We therefore consider the extent to which collider bias could be influencing the MR results due to a healthy participant effect to be small. Finally, despite undertaking sensitivity analyses, we cannot exclude the possibility that our findings may be driven by vertical pleiotropic effects of the genetic instruments. As the metabolome is comprised of thousands of metabolites connected through numerous reactions, some of which may be unknown, the distinction between horizontal and vertical pleiotropy is difficult to make in the context of metabolomics. The 2 SNP score, comprised of loci in genes of major enzymes of glycine catabolism, is the most specific score to the glycine pathway but still shows modest associations with some metabolites of which we cannot be sure that they are metabolically linked to glycine, e.g., the unknown metabolite X-16570, and with glycine-conjugated metabolites, which may have effects independent of glycine on cardio-metabolic disease risk. Therefore, we cannot fully exclude that glycine-independent mechanisms may have biased the genetically

predicted association of glycine with disease risk. Moreover, as glycine is a metabolite on the intersection of many metabolic pathways, the genetic association of glycine with cardio-metabolic diseases may represent the causal effect of a metabolite to which glycine is metabolically close linked (e.g., tetrahydrofolate or serine).

In conclusion, we show that low glycine is associated with higher incidence of CHD and that genetic scores for glycine are compatible with this relationship being causal. We furthermore show that a glycine-lowering effect of IR may drive the consistently reported association between higher glycine and lower incidence of T2D, while the evidence for a causal relationship between glycine and T2D risk is weaker.

## Methods

**Studies**. The EPIC-Norfolk study is a cohort of 25,000 individuals aged between 45 and 74 from the general population of Norfolk (East England)[57], nested within the European Prospective Investigation into Cancer and Nutrition (EPIC). The study was approved by the Norfolk Research Ethics Committee (ref. 05/Q0101/191) and all participants gave their written consent before entering the study. Untargeted metabolomics measurements using the DiscoveryHD4® platform[58] (Metabolon, Inc., Durham, USA) on non-fasted plasma samples have been completed in a T2D case-cohort study which included all 586 incident T2D cases and 746 quasi randomly selected participants from the entire cohort, and two sub-cohorts of quasi randomly selected participants—5989 participants in sub-cohort A and 5977 participants in sub-cohort B. There was no overlap between the T2D case-cohort and sub-cohorts A and B. Further details about the metabolomics measurements and QC have previously been described elsewhere[59]. Genome-wide genotyping was done using the Affymetrix UK Biobank Axiom array and genotype data were imputed to the 1000 Genomes Phase 3 reference panel[60] using IMPUTE2[61]. Data from EPIC-Norfolk were used to conduct a GWAS for glycine (sub-cohort of the T2D case-cohort and sub-cohort A), to estimate sex-specific effect sizes of the lead SNPs for glycine on glycine levels (sub-cohort A and B), to test associations of the genetic scores and variants for glycine across the metabolome (sub-cohorts A and B), and to assess the observational association of glycine levels with incident CHD, MI, stroke (sub-cohorts A and B) and T2D (T2D case-cohort).

EPIC-CVD, another sub-study of EPIC, is a prospective case-cohort study focussing on the risk factors of cardiovascular diseases and includes nearly 14,000 incident CHD cases and a sub-cohort comprised of 18,249 randomly selected participants[62]. Ethical approval was obtained from the ethics committees of the International Agency for Research on Cancer and the local institutions where the participants were recruited. Genotyping was performed using the HumanCoreExome array, the Quad 660 array and the Infinium OmniExpressExome array (all from Illumina) and genotyped data were imputed based on the Haplotype Reference Consortium reference panel. Data on 28,217 participants (33.7% CHD cases) of the EPIC-CVD study were used to assess the effect sizes of the genetic variants for glycine with CHD, for both sexes combined and separately.

InterAct is a T2D case-cohort study nested within EPIC, which was designed to study the interaction between lifestyle and genetic factors in relation to T2D[63]. All participants gave written informed consent and ethical approval was given by the ethics committees of the International Agency for Research on Cancer and the local institutions. The study includes 12,403 incident cases of T2D and a randomly selected sub-cohort of 16,154 individuals from nine European countries. Genome-wide genotyping was done using the HumanCoreExome array. Genome-wide genotyped data were imputed to the European 1000 Genomes reference panel (March, 2012 release) using IMPUTE2. Data from the InterAct study were used to assess the sex-specific associations of the genetic loci for glycine with risk of T2D.

The Fenland study is a longitudinal cohort study including more than 12,400 participants born between 1950 and 1975 from the general population of Cambridgeshire (UK)[59]. Ethical approval for the study was was given by the Cambridge Local Ethics committee (ref. 04/Q0108/19) and all participants gave their written consent prior to entering the study. Fasted plasma concentrations of 174 metabolites were measured using the AbsoluteIDQ® p180 Kit (Biocrates Life Sciences, Innsbruck, Austria). Further details about the metabolomics measurements and QC have previously been described elsewhere[59]. Genome-wide genotyping of the Fenland participants was done in two waves; the first 1400 individuals were genotyped using the Affymetrix Genome-Wide Human SNP Array 5.0 and the next 9369 participants on the Affymetrix UK Biobank Axiom Array. The same imputation strategy as for the EPIC-Norfolk study was used. GWAS for glycine and assessment of the sex-specific effect sizes of the glycine variants were run in the Fenland study.

The INTERVAL study is a randomised trial of ~50,000 whole blood donors enrolled from all 25 static centres of NHS Blood and Transplant[64]. All participants gave written informed consent and the study was approved by NRES Committee East of England - Cambridge East (ref. 11/EE/0538). For the present study, non-fasting serum blood samples were provided by unrelated individuals from the

INTERVAL trial. The samples were analysed using a high-throughput serum NMR metabolomics platform[65,66], which provided information on 230 metabolites, including glycine. We removed participants with >30% of metabolite measures missing, duplicated individuals, and metabolic data more than 10 SDs from the mean. Genotyping was conducted using the Affymetrix UK Biobank Axiom array. Prior to imputation, SNPs missing in more than 1% of the samples or failing in more than one batch were excluded. Monomorphic and multi-allelic variants and variants that failed to pass the threshold on clustering quality, were intensity outliers or deviated from Hardy-Weinberg equilibrium were omitted. The data were imputed to a joint 1000 Genomes Phase 3 (May 2013)-UK10K reference imputation panel. Data from 40,509 INTERVAL participants with glycine measures were used to run a GWAS for glycine and to assess the sex-specific effect sizes of the glycine variants on glycine.

The UK Biobank study is a longitudinal cohort study of more than 500,000 participants from across the UK[67]. All participants gave their informed consent. Genotyping was performed using the UK Biobank Axiom Array and imputation was based on the reference panel from the Haplotype Reference Consortium, using IMPUTE2. Associations of the genetic variants for glycine with CHD, T2D and blood pressure were assessed in the UK Biobank.

**Genome-wide association analyses for glycine**. GWAS for glycine levels were conducted in the Fenland ($n = 9324$), EPIC-Norfolk ($n = 5840$) and INTERVAL ($n = 40,509$) studies. Glycine levels were natural log transformed, winsorised at 5 SDs and transformed to the $Z$ score. Analyses were conducted based on a generalised linear mixed model adjusted for age, sex and the first four principal components, using BOLT-LMM[68]. For the Fenland study, separate GWAS were run for the two different genotyping arrays, which were meta-analysed using a fixed-effects model in METAL[68]. The GWAS in EPIC-Norfolk only included the sub-cohort of the T2D case-cohort study and sub-cohort A, as at the time the GWAS was conducted, metabolite measurements in sub-cohort B were not yet available. For the GWAS in EPIC-Norfolk, batch (i.e., sub-cohort of T2D case-cohort or sub-cohort A) was included as an additional covariate. Genetic variants were excluded if the standard error (SE) > 10 or <0, the absolute value of beta >5, $p$ for the Hardy-Weinberg equilibrium $<1 \times 10^{-6}$ or info score <0.4 (INTERVAL) or <0.3 (Fenland and EPIC-Norfolk).

**Meta-analyses of GWAS for glycine**. Results of the Fenland, EPIC-Norfolk and INTERVAL studies were meta-analysed with publicly available summary-level GWAS results from an NMR-based multi-cohort discovery by Kettunen et al.[11] (downloaded from http://www.computationalmedicine.fi/data#NMR_GWAS) and from TwinsUK published by Shin et al.[69] (downloaded from http://mips.helmholtz-muenchen.de/proj/GWAS/gwas/index.php?task=download), resulting in a total sample size of up to 80,003 individuals. Meta-analysis of the five studies was conducted using METAL[70] and was based on the $p$-values, directions of effect and sample sizes, to minimize the effect of heterogeneity due to differences between metabolomics platforms and in analytical decisions between the studies, such as transformation of the glycine measures and inclusion of covariates (Supplementary Table 1). As the five studies differed in terms of adopted imputation strategy, analyses were restricted to variants present in at least two studies and >50% of the total sample size, and with MAF ≥0.1%. Pooled effect sizes and SEs were generated through an effect size-based meta-analysis of the Fenland, EPIC-Norfolk (sub-cohort of T2D case-cohort and sub-cohort A) and INTERVAL studies (total $n = 55,673$) in METAL.

**Identification of primary and secondary genetic signals**. Distance-based clumping using 1 Mb windows was used to identify independent loci significantly associated with glycine ($p < 5 \times 10^{-8}$). Because of the very strong and wide signal at *CPS1*, a window of 3 Mb on both sides of the lead variant rs715 was used to capture the entire locus. Secondary signals were identified through approximate conditional analyses using GCTA-COJO[71]. To maximise the sample size, this analysis was conducted on the pooled $Z$ scores generated in the $p$-value-based meta-GWAS. Variants with MAF < 1% were omitted from the conditional analyses. HRC-imputed genome-wide data from 19,318 EPIC-Norfolk participants were used as an LD reference panel and a joint $p$-value threshold of $5 \times 10^{-8}$ was used to identify secondary signals. Of the 73 variants selected through the approximate conditional analysis, 11 were in LD ($r^2 > 0.05$) with another selected variant and were therefore omitted. At the *CPS1* locus, 44 secondary signals remained of which 27 were low-frequency variants. A second LD filter to remove variants in high LD with the common sentinel variant ($D' < 0.05$) was applied, after which only five variants at the *CPS1* locus remained.

**Sex-specific effect sizes of glycine loci**. Sex-specific effect sizes for lead SNPs at the 27 loci on glycine levels standardised by sex were estimated in the INTERVAL, Fenland and EPIC-Norfolk studies (sub-cohorts A and B) (in total 30,226 men and 31,957 women) and meta-analysed using the R package 'metafor'[72]. As a sensitivity analysis, sex-specific effect sizes on natural-log transformed instead of within-sex standardised glycine levels were estimated in the Fenland and EPIC-Norfolk studies (9927 men and 11,284 women).

**Annotation of lead SNPs and genetic scores for glycine**. Lead variants at the 27 loci identified for glycine levels were annotated using ANNOVAR[73] and we searched for published associations for all lead variants and variants in high LD ($r^2 > 0.6$) using the PhenoScanner web browser[74]. Reported associations with metabolites were identified using SNiPA for the lead variants and variants within 1 Mb of or in high LD ($r^2 > 0.8$) with the lead variants[75].

Associations of the four different glycine scores and rs715 at *CPS1* with 894 metabolites measured in sub-cohorts A and B and at least 50% of the total sample size of the EPIC-Norfolk study were assessed. Metabolite levels were natural log-transformed and transformed to the $Z$ scores and models were adjusted for sex and metabolite batch.

**Associations of glycine lead SNPs with diseases and traits**. Sex-combined and sex-specific effect estimates of the lead variants for glycine (or a variant in high LD ($r^2 > 0.7$) if the lead variant was not available) on CHD were assessed in the EPIC-CVD study and the UK Biobank, and obtained as lookups from CARDIo-GRAMplusC4D[25] for the sex-combined associations (60,801 cases and 123,504 controls, downloaded from http://www.cardiogramplusc4d.org/data-downloads/) and from the German MI family studies[25,76–78] for sex-specific associations (994 female and 2804 male cases; 2752 female and 2554 male controls). The associations of the genetic variants for glycine with CHD were tested in EPIC-CVD based on Cox proportional hazards models and using Prentice weighting and robust standard errors (3712 female and 5786 male cases; 14,764 female and 13,453 male controls). Models were adjusted for age, genotyping array, testing centre, the first four genetic PCs and sex for the sex-combined analyses. In the UK Biobank, Cox proportional hazards models adjusted for age, the first 10 PCs, genotyping chip and sex (for sex-combined analyses) were run on 18,501 incident and prevalent CHD cases (5147 women) and 333,545 controls (184,608 women).

Associations of the glycine SNPs with any stroke and ischemic stroke were based on look-ups in the summary-level GWAS results from the MEGASTROKE consortium[79] (downloaded from http://www.megastroke.org/download.html) and associations in the UK Biobank (Any stroke: up to 48,916 cases and 765,017 non-cases; ischemic stroke: up to 37,771 cases and 764,290 non-cases). Associations with haemorrhagic stroke were based on data from UK Biobank only (1655 cases and 365,988 non-cases). Stroke and stroke sub-type cases in UK Biobank were identified based on ICD-9 and ICD-10 codes, cause of death (based on ICD-10 codes) and on self-reported diagnoses based on a verbal interview.

Associations of the glycine variants with T2D for sexes combined were obtained from the latest GWAS on T2D by the DIAGRAM consortium on 74,124 cases and 824,006 controls[80] (downloaded from http://www.diagram-consortium.org/downloads.html). Sex-specific associations with T2D were based on the InterAct (4712 female and 4596 male cases; 7190 female and 4333 male controls) and UK Biobank studies (7301 female and 12,318 male cases; 181,442 female and 149,249 male controls). Sex-specific GWAS for T2D in the InterAct study were run using SNPTEST, based on logistic regression models adjusted for age, assessment centre and the first 10 genetic PCs. In the UK Biobank, associations of the SNP dosages with prevalent and incident T2D, identified based on ICD10 codes, were estimated by fitting logistic regression models adjusted for age, four genetic PCs and genotyping chip in STATA v15.0 (StataCorp, College Station, Texas, USA). Only UK Biobank participants from the subset of unrelated British ancestry participants were included. Meta-analyses of the sex-specific effect sizes were meta-analysed using the R package 'metafor'[72].

The effects of the glycine variants on SBP and DBP were assessed using data on 445,360 UK Biobank participants (241,417 women) of European ancestry. GWAS on rank-based inverse normally transformed SBP and DBP were conducted within sex and using BOLT-LMM. For individuals who reported to be taking antihypertensive medication, 15 mmHg and 10 mmHg was added to measured SBP and DBP[81]. Analyses were adjusted for age, age[2], BMI and genotyping array. For variants both genotyped and imputed, imputed probabilities were used if the variant was imputed well (INFO > 0.7) and the genotyping call rate was less than 98%. Sex-combined estimates were generated through fixed-effect meta-analyses of the sex-specific estimates.

Associations of the glycine variants with triglycerides, LDL, HDL and total cholesterol were obtained from publicly available GWAS summary-results based on up to 188,577 participants from the Global Lipids Genetics Consortium[16] (downloaded from http: //csg.sph.umich.edu/abecasis/public/lipids2013/). Look-ups for 13 blood cell traits came from publicly available GWAS summary results based on a genetic discovery in 173,480 participants[82] (downloaded from http://www.bloodcellgenetics.org/).

Look-ups for overall breast cancer risk were obtained from a European ancestry genetic discovery for breast cancer including up to 122,977 cases and 105,974 controls[26] (downloaded from http://bcac.ccge.medschl.cam.ac.uk/bcacdata/oncoarray/gwas-icogs-and-oncoarray-summary-results/). Genetic associations with epithelial ovarian cancer were obtained from publicly available results of a genetic discovery including 25,509 cases and 40,941 controls of European ancestry from OCAC[27] (downloaded from http://ocac.ccge.medschl.cam.ac.uk/data-projects/results-lookup-by-region/). Look-ups for prostate cancer came from the publicly available summary-level results of a GWAS including 79,148 prostate cancer cases and 61,106 controls of European ancestry (downloaded from: http://practical.icr.ac.uk/blog/?page_id=8164).

**MR methods**. MR analyses for glycine to cardio-metabolic diseases and cancers were based on summary-level data and four different methods: inverse variance-weighted MR[83], MR-Egger[53], weighted median MR and penalised weighted median MR[52]. Heterogeneity in the effects of the genetic variants on the outcome was assessed based on the Cochran's Q statistic, and directional pleiotropy was estimated based on the MR-Egger intercept. The weighted median MR was used for the main analyses, as across the analyses and phenotypes, there was proof of heterogeneity ($p$ for Cochran's Q $< 0.05$ and/or $p$ for Egger's intercept $< 0.05$).

Two-sample summary-level data multivariable MR analyses to obtain the effect size of glycine levels adjusted for blood pressure on CHD were run as previously described by Day et al.[84]. In brief, weighted multilinear models were fitted with the effect sizes of the glycine SNPs on glycine and on the SBP and/or DBP as the explanatory variables and the effect sizes of the glycine SNPs on CHD as the independent variable. Weighting was based on the inverse variance of the effect sizes of the genetic variants for glycine on CHD.

Reverse MR analyses were conducted to assess the causality of risk factors for T2D on glycine levels using the same analytical approach as for the forward MR analyses. Previously published and validated genetic scores by the GIANT and MAGIC consortia were used for BMI[85] (97 genetic variants), fasting insulin adjusted for BMI as a marker of IR[86] (10 genetic variants) and insulin levels at 30 min during an oral glucose tolerance test as a marker for early-phase IS[86,87] (21 genetic variants). As there was no evidence for heterogeneity or pleiotropy, the inverse variance-weighted MR method was used.

**Observational analyses**. We tested if glycine levels at baseline were associated with incidence of CHD, MI and stroke in the EPIC-Norfolk study. Analyses for CHD, MI and stroke were performed in sub-cohorts A and B, while analyses for T2D were conducted in the T2D case-cohort study (586 incident cases and 746 quasi randomly selected participants). During follow-up, 2053 participants developed CHD, of which 659 participants had MI. In total, 1163 participants had a stroke, of which 212 cases were confirmed to be haemorrhagic and 569 were confirmed to be ischemic. Cox proportional hazards models were fitted to estimate the HRs for disease incidence based on 1 SD increase in glycine levels. Age at recruitment was used as the underlying timescale, and models were adjusted for sex, and additionally for BMI, waist-hip ratio, educational attainment, smoking, alcohol consumption, physical activity, blood pressure and blood lipids in the fully adjusted models. Prevalent disease cases were excluded from the analyses. Prentice weighting was applied to the models for T2D, to adjust for the enrichment of cases. Analyses for CHD, stroke and MI were conducted for the sub-cohorts separately and meta-analysed using metafor[72]. Sex-specific analyses were conducted on within-sex standardised glycine levels.

The association of glycine levels with sex, age, smoking (number of cigarettes/ day), alcohol consumption (grams of alcohol/day) and physical activity (energy expenditure from physical activity/day) was tested based on a linear regression model adjusted for metabolomics batch and included 10,475 participants (5612 women) of the Fenland study.

**Reporting summary**. Further information on experimental design is available in the Nature Research Reporting Summary linked to this article.

## Data availability

The summary statistics of the meta-GWAS for glycine levels are available as Supplementary Data 5. All other data are contained within the article and its supplementary information, or can be obtained upon reasonable request from the corresponding author.

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

## Acknowledgements

We are grateful to participants and staff for their contribution to the studies included in this project. The Fenland Study is supported by the UK Medical Research Council (MC_UU_12015/1 and MC_PC_13046). We thank Dr Larissa Richardson and Dr Luke Marney for the acquisition of the metabolomics data. EPIC-Norfolk has received funding from the Medical Research Council (MR/N003284/1 and MR/L00002/1), Cancer Research United Kingdom (14136) and the Innovative Medicines Initiative Joint Undertaking under EMIF grant agreement number 115372 (European Union's Seventh Framework Programme FP7/2007-2013) and EFPIA companies' in kind contribution. The EPIC-InterAct study was funded by the EU FP6 Programme (LSHM_CT_2006_037197). We particularly thank Nicola Kerrison (EPIC-InterAct Data Manager). EPIC-CVD has been supported by the European Union Framework 7 (HEALTH-F2-2012-279233), the European Research Council (268834), the UK Medical Research Council (G0800270 and MR/L003120/1), the British Heart Foundation (SP/09/002 and RG/08/014 and RG13/13/30194) and the UK National Institute of Health Research (NIHR). The coordination of EPIC is financially supported by the European Commission (DG-SANCO) and the International Agency for Research on Cancer. We thank Sarah Spackman (EPIC-CVD Data Manager). INTERVAL: Participants were recruited with the active collaboration of NHS Blood and Transplant England, which has supported field work and other elements of the trial. This study was funded by the NIHR, the NIHR BioResource, the NIHR [Cambridge Biomedical Research Centre at the Cambridge University Hospitals NHS Foundation Trust] [*], the European Commission Framework Programme 7 (HEALTH-F2-2012-279233), the NIHR Blood and Transplant Research Unit in Donor Health and Genomics (NIHR BTRU-2014-10024), UK Medical Research Council (MR/L003120/1) and the British Heart Foundation (RG/13/13/30194). *The views expressed are those of the author(s) and not necessarily those of the NHS, the NIHR or the Department of Health and Social Care.
UK Biobank: This research has been conducted using the UK Biobank resource under Application Numbers 12885 and 20480. Additional acknowledgements are provided in Supplementary Note 1.

## Author contributions

L.B.L.W. and C.L. conceptualised the project and interpreted the results, with input from L.A.L., A.S.B., J.M.M.H and S.B. Genome-wide analyses for glycine were conducted by I.D.S., L.A.L., L.B.L.W. and C.O.-W., L.B.L.W. constructed and validated the genetic scores for glycine and conducted the MR analyses. Associations of genetic variants in UK Biobank were assessed by L.A.L. for CHD and T2D, by S.B. for stroke and by J.M.M.H., P.S. and S.K. for blood pressure. Observational analyses were conducted by L.B.L.W. and L.A.L. The manuscript was written by L.B.L.W., with input from all co-authors.

## Additional information

**Competing interests:** R.A.S. is an employee and shareholder of GlaxoSmithKline. C.O.-W. received prize money from Novartis (<£1000). J.D. reports grants and personal fees from Merck Sharp & Dohme (MSD), grants and personal fees from Novartis, grants from British Heart Foundation, grants from European Research Council, grants from NIHR, grants from NHS Blood and Transplant, grants from Pfizer, grants from UK MRC, grants from Wellcome Trust, grants from AstraZeneca, outside the submitted work. J.D. also reports non-financial support from Merck Sharp & Dohme (MSD) and non-financial support from Novartis, outside the submitted work. A.B. has received grants outside of this work from AstraZeneca, Biogen, Merck, Novartis and Pfizer and personal fees outside of this work from Novartis. The remaining authors declare no competing interests.

