## [Peer Review File · Nature Communications]

Reviewer #1 (Remarks to the Author):

Assessing the causal association of glycine with risk of coronary heart disease

Wittebans et al.

This manuscript describes a large-scale meta-analysis of plasma glycine levels in >80K subjects and the identification of 22 novel loci. The authors then use the variants at these loci to infer causality between glycine and CHD as well as to investigate potential biological mechanisms. Their results indicate that glycine levels are inversely associated with risk of CHD but that this protective association is likely not sex-specific. Furthermore, the data suggest that glycine is associated with reduced risk of CHD through mechanisms related to blood pressure. Overall, the study is carried out well, and the results are straightforward and self-explanatory. However, there are certain points the authors may wish to address as outlined below:

1. Is this meta-analysis restricted to HAPMAP-based imputed SNPs since, at least to the knowledge of this reviewer, 1000G imputed metabolomics GWAS results for TwinsUK and KORA are not publicly available? The authors should make this point clear in their results and methods as it is not currently indicated. This would presumably decrease the number of loci that could be identified, especially with the large number of subjects used in this study. For example, there is evidence that rare variants can also influence glycine levels (see Human Molecular Genetics, 2018, Vol. 27, No. 9 1664–1674). These limitations should be addressed in the discussion.
2. The authors used a Z-score method for their meta-analysis given the different platforms on which metabolomics was carried out in the different cohorts. It would be of interest to show the glycine associations for the 27 loci separately in the cohorts on which absolute quantification was done (ie Biocrates) vs. relative quantification (ie Metabolon) and whether there are differences between platforms. This can be provided in the supplement.
3. glycine has a relatively short half life and its levels could potentially be affected by fasting/nonfasting status. Therefore, the authors should indicate whether glycine levels were different between cohorts where non-fasting plasma vs plasma was used for metabolomics measurements and also should show that associations at the 27 loci did not differ between these cohorts.

4. The prior meta-analysis by Draisma et al included TwinsUK and KORA so the authors should clarify whether the subjects from these two cohorts are the same as those in the meta-analysis by Shin et al. Otherwise, it would be seem that TwinsUK and KORA are being included twice in the current meta-analysis?

5. The authors argue based on the results of the 2-SNP glycine-specific score that this amino acid is the putative causal and CHD-protective metabolite. The data are supportive of this assertion but until directly proven in animal models or a clinical trial, the authors should address the fact that even the 2-SNP glycine-specific score is still associated with unknown metabolites (ie X-16570 and X-13722). Thus, it is still possible that these metabolites could also be casually related to CHD risk, even though the effect sizes are weaker. Notably, rs715 is also associated with these unknowns. This should be addressed in the discussion (line 331).

6. The authors should provide sex-stratified results for rs715 with SBP and DBP in Fig 6 as well. If the protective effect of glycine is mediated through blood pressure, then the association of the CPS1 variant with these traits should mimic all other dimorphic clinical trait associations reported for this locus.

7. The authors state that they had low power to quantify the extent to which the genetic association of glycine levels with CHD risk could be explained by blood pressure. However, perhaps they can still assess this based on known quantitative epidemiological associations between BP and CHD. Assuming this is possible, does the effect of the 5- or 2-SNP glycine GRS on BP explain all of the 20% decreased risk of CHD associated with both these GRS? In other words, could there still be residual protective effects of glycine independent of BP?

8. There is no indication that the GWAS results from this meta-analysis will be made publicly available for download by other investigators. The authors should do this in the same fashion that allowed them to use summary level GWAS data from TwinsUK, KORA, and the Magnetic consortium.

Reviewer #2 (Remarks to the Author):

Wittebans et al have used large epidemiological cohorts to identify genetic determinants of plasma glycine, to assess the possible causal role of glycine in CHD risk, and to examine the biological pathways underlying the possible glycine-CHD association. They found 27 genetic regions, of which 22 were novel, significantly associated with plasma glycine levels. High glycine levels were associated with lower incidence of CHD and the Mendelian randomization analyses suggested that this association was causal. Furthermore, higher glycine levels were associated with lower systolic and diastolic blood pressure and the authors conclude that the inverse relationship between glycine and CHD risk is driven by blood pressure.

The study is based on large data sets and succeeds to uncover some novel biology. Generally, the paper is interesting and well-written, the methods seem appropriate and the conclusions are mostly supported by the data. However, some interpretations can be questioned and in some points the presentation could be clearer and more balanced. My comments are as follows:

1) The effect of glycine on CHD risk looks quite modest (OR =0.95 per one SD) and after adjusting for BP it disappears totally. I understand that the authors have focused on discovering novel biology here and not aimed at CHD risk prediction but nevertheless a brief comment on clinical significance of these findings would be appropriate.

2) The authors seem to have ignored totally the association of glycine with incident diabetes. A recent report on Framingham study participants with normal fasting glucose at baseline described an inverse association of plasma glycine with the risk of incident diabetes. The Framingham authors also carried out a MR analysis and argue for a causal relationship between glycine levels and the risk of incident DM. (Merino J et al. *Diabetologia*. 2018 Jun;61(6):1315-1324). Clearly, the authors should consider DM in the present analyses and comment on the paper of Merino et al.

3) If the effect of glycine on CHD risk is mediated through blood pressure as the authors argue, it is a bit strange that no association was observed between glycine and the risk of stroke. Usually, blood pressure is a stronger risk factor for stroke than for CHD and one would expect also here that the life-long lower BP level in persons with high glycine would be reflected as lower risk of stroke.

4) Glycine was measured in different laboratories with different methods and different fasting times: Metabolon mass-spec from non-fasted plasma samples in the EPIC-Norfolk study; AbsoluteIDQ p180 Kit of Biocrates (i.e. mass spec) from fasted plasma samples of the Fenland study; and NMR metabolomics from non-fasting serum samples of the INTERVAL trial. Were any overlapping measurements performed? Do we know anything about the agreement of glycine measurements in different conditions, i.e., plasma vs. serum, fasting vs. non-fasting, and the two different mass-spec platforms vs. the NMR platform? There is no common standardization in the metabolomics field and this raises a concern that the results can vary considerably depending on platform and other measurement conditions.

5) The largest material of the study comes from the UK Biobank, which has a participation rate of about 6% and the participants are known to be healthier and socioeconomically better off than the average UK population. The second largest material, the INTERVAL trial consists of 50,000 blood donors, who also are likely to be healthier than the average population. The authors should at least comment on whether the "healthy participant effect" has any influence on these results.

Minor comments:

- 1) If glycine is causally related to CHD risk, it would be of interest to also analyze the “environmental” determinants of glycine levels and show for comparison the proportion of variance they explain. This should be possible in at least some of the large data sets the authors have.
- 2) Fig. 6 and supplementary Figs 1 and 4: It would be clearer to have a vertical line at 0 in these figures.
- 3) How did you take antihypertensive medications into account in the analyses on BP?
- 4) P. 22, Cox PH reg modelling: Please describe the models more clearly. Did you exclude persons with a history of CHD or stroke at baseline, or are they included in these models? Were age and sex the only covariates? What about DM, smoking, lipids?

Reviewers' comments:

Reviewer #1 (Remarks to the Author):

Assessing the causal association of glycine with risk of coronary heart disease

Wittemans et al.

This manuscript describes a large-scale meta-analysis of plasma glycine levels in >80K subjects and the identification of 22 novel loci. The authors then use the variants at these loci to infer causality between glycine and CHD as well as to investigate potential biological mechanisms. Their results indicate that glycine levels are inversely associated with risk of CHD but that this protective association is likely not sex-specific. Furthermore, the data suggest that glycine is associated with reduced risk of CHD through mechanisms related to blood pressure. Overall, the study is carried out well, and the results are straightforward and self-explanatory. However, there are certain points the authors may wish to address as outlined below:

1. Is this meta-analysis restricted to HAPMAP-based imputed SNPs since, at least to the knowledge of this reviewer, 1000G imputed metabolomics GWAS results for TwinsUK and KORA are not publicly available? The authors should make this point clear in their results and methods as it is not currently indicated. This would presumably decrease the number of loci that could be identified, especially with the large number of subjects used in this study. For example, there is evidence that rare variants can also influence glycine levels (see Human Molecular Genetics, 2018, Vol. 27, No. 9 1664–1674). These limitations should be addressed in the discussion.

Thank you, we agree that our description was not sufficiently clear and we have now improved and extended this in the methods (lines 547-561), discussion (lines 429-433) and Supplementary Table .1 In brief, analyses were not limited to variants covered on HapMap2, but based on imputation to 1000G, HRC and UK10K for over 93% of the overall sample size (80k participants), with the remainder being restricted to HapMap2 (publicly available data from TwinsUK samples, Shin *et al.*), and therefore include 10,249,597 high quality genotyped and imputed variants for the vast majority of the sample.

We absolutely agree that rare variants contribute. We have now included the important and very relevant reference highlighted by the reviewer (lines 57 and 323), and also state more clearly that 2 of the 27 genome-wide significant loci have a MAF<1% (discussion lines 321-323). We also clarify more explicitly that we were unable to include the 3 identified low-frequency or

rare variants in the MR analyses as these were not covered in publicly available GWAS datasets. In addition, they are not expected to meaningfully contribute to the MR estimates due to their low weight in the analysis.

2. The authors used a Z-score method for their meta-analysis given the different platforms on which metabolomics was carried out in the different cohorts. It would be of interest to show the glycine associations for the 27 loci separately in the cohorts on which absolute quantification was done (ie Biocrates) vs. relative quantification (ie Metabolon) and whether there are differences between platforms. This can be provided in the supplement.

We agree with the reviewer on the importance of assessing between-platform differences in genetic effect sizes and have addressed this issue now in the manuscript, to the extent this was possible with the data available to us, as follows:

- 1) We tested for heterogeneity between the 5 studies included in the Z score-based meta-analysis of GWAS for glycine (Supplementary Table 2 column J). No heterogeneity was observed, except for the *CPS1* locus (rs715) (lines 83-84). Because the *CPS1* locus has an unusually high significance ($p=3 \times 10^{-1632}$) and very strong effect size (per-allele beta on standard deviations of glycine=0.444), we think that the identified heterogeneity for the *CPS1* locus may have reached significance despite relatively small differences in effect sizes between the 3 studies for which the effect sizes were on the same scale (Supplementary Figure 1).
- 2) We tested for heterogeneity between the 3 studies and platforms for which the effect sizes were on the same scale: Fenland (Biocrates p180), EPIC-Norfolk (Metabolon Discovery HD4) and INTERVAL (NMR platform) (see Supplementary Table 2, Column M). With the exception of the *CPS1* locus, no evidence for heterogeneity in the effect sizes between the 3 studies was found. We also included forest plots of the effect sizes of the 27 loci in the 3 studies (Supplementary Figure 1).
- 3) We could not conduct a more complete comparison of the effect sizes between the platforms, because the effect sizes from the publicly available GWAS results are based on a different scale and/or phenotype transformation than the 3 in-house studies. We applied consecutively natural-log transformation, winsorisation to 5 SDs and Z score transformation to glycine levels in the Fenland, EPIC-Norfolk and INTERVAL studies. Conversely, Shin *et al.* only applied base 10 log-transformations to metabolite levels, while a rank-based inverse normal transformation was applied to the metabolite levels by Kettunen *et al.*

3. Glycine has a relatively short half life and its levels could potentially be affected by fasting/nonfasting status. Therefore, the authors should indicate whether glycine levels were different between cohorts where non-fasting plasma vs plasma was used for metabolomics measurements and also should show that associations at the 27 loci did not differ between these cohorts.

We agree and expect, as the reviewer indicates, that glycine levels may be affected by fasting status. Gannon *et al.* (The American Journal of Clinical Nutrition, 2002, Volume 76, Issue 6, pages 1302-1307) showed that, when fasted participants were given an oral dose of glycine, the plasma concentration of glycine sharply increased, with the maximum concentration measured at 40 min after administration, after which glycine concentrations gradually decreased over the next few hours to the fasted concentration. As glycine is a very common amino acid in dietary protein, we expect that the plasma concentration of glycine will be lower in fasted than non-fasted individuals. We however do not have data available to us that allow us to confirm this hypothesis. We tested for differences in effects of genetic loci, as proposed, and found no evidence that genetic effects on glycine differed between studies with fasted versus non-fasted samples, except for *CPS1* (see Supplementary Table 2 and response to previous comment).

4. The prior meta-analysis by Draisma et al included TwinsUK and KORA so the authors should clarify whether the subjects from these two cohorts are the same as those in the meta-analysis by Shin et al. Otherwise, it would be seem that TwinsUK and KORA are being included twice in the current meta-analysis?

Thank you for bringing this to our attention. To avoid including KORA twice, we had restricted inclusion of publicly available GWAS results to Kettunen *et al.* and TwinsUK but not KORA from Shin *et al.* GWAS results by Draisma *et al.* were not included. We have clarified this in the description of the methods (lines 549-553) and added Supplementary Table 1 which gives an overview of the included studies and sample sizes.

5. The authors argue based on the results of the 2-SNP glycine-specific score that this amino acid is the putative causal and CHD-protective metabolite. The data are supportive of this assertion but until directly proven in animal models or a clinical trial, the authors should address the fact that even the 2-SNP glycine-specific score is still associated with unknown metabolites (ie X-16570 and X-13722). Thus, it is still possible that these metabolites could also be casually related to CHD risk, even though the effect sizes are

weaker. Notably, rs715 is also associated with these unknowns. This should be addressed in the discussion (line 331).

We agree with the reviewer and acknowledge that, while loci near *GLDC* and *GCSH* are excellent candidate instruments for glycine because of their direct biological link to glycine catabolism, the 2 SNP score reaches significance for 7 other metabolites than glycine, which may indicate some degree of pleiotropy. However, these metabolites were mostly metabolically linked to and/or observationally correlated with glycine.

Based on the mass spec fragmentation pattern, Metabolon expects that the unknown metabolite X-16570, which is the only unknown metabolite reaching significance for the 2 SNP score, to be a glycine-conjugated fatty acid. This still requires confirmation based on a chemical standard, so we can at this point not exclude that the modest association with X-16570 may indicate some pleiotropy. We have now added a statement outlining this in more detail in the discussion (lines 442-452).

6. The authors should provide sex-stratified results for rs715 with SBP and DBP in Fig 6 as well. If the protective effect of glycine is mediated through blood pressure, then the association of the *CPS1* variant with these traits should mimic all other dimorphic clinical trait associations reported for this locus.

We do indeed find a similar sex difference in the associations of rs715 with SBP and DBP as we found for glycine, with stronger effect sizes for women than for men. As suggested by the reviewer, we have now included the sex-specific associations of rs715 with SBP and DBP in Supplementary Figure 6 and refer to this in the main text on lines 236-237.

7. The authors state that they had low power to quantify the extent to which the genetic association of glycine levels with CHD risk could be explained by blood pressure. However, perhaps they can still assess this based on known quantitative epidemiological associations between BP and CHD. Assuming this is possible, does the effect of the 5- or 2-SNP glycine GRS on BP explain all of the 20% decreased risk of CHD associated with both these GRS? In other words, could there still be residual protective effects of glycine independent of BP?

Apologies if we did not make it clear enough what was tested. We did indeed investigate whether the genetic association of glycine with CHD can be explained by SBP and DBP based on the 5 SNP score, as proposed and found that adjustment for SBP, DBP or both accounted for most or all, respectively, of the genetic effect of glycine on CHD. However, because, confidence intervals of these analyses were wide, we want to remain cautious about blood pressure being the main mediator of this observation (lines 345-350). In addition, the lack of association

between glycine and stroke suggests that blood pressure is unlikely to be the only mediating pathway between glycine and disease risk (lines 348-350).

8. There is no indication that the GWAS results from this meta-analysis will be made publicly available for download by other investigators. The authors should do this in the same fashion that allowed them to use summary level GWAS data from TwinsUK, KORA, and the Magnetic consortium.

We fully agree about the importance of making summary-level GWAS data accessible to the research community. We are in the final stages of finishing a large multi-platform genetic discovery project for all metabolites covered on the Biocrates p180 platform, which includes the genetic discovery for glycine presented in this manuscript, and are setting up a publicly available web server on which the summary-level GWAS data for all metabolites can be browsed, visualised and downloaded. Until this web server becomes online, data access to the summary-level GWAS results will be given to other researchers upon simple request by email to the corresponding author. We have included a section on data availability at the end of the manuscript (lines 702-704).

Reviewer #2 (Remarks to the Author):

Wittebans et al have used large epidemiological cohorts to identify genetic determinants of plasma glycine, to assess the possible causal role of glycine in CHD risk, and to examine the biological pathways underlying the possible glycine-CHD association. They found 27 genetic regions, of which 22 were novel, significantly associated with plasma glycine levels. High glycine levels were associated with lower incidence of CHD and the Mendelian randomization analyses suggested that this association was causal. Furthermore, higher glycine levels were associated with lower systolic and diastolic blood pressure and the authors conclude that the inverse relationship between glycine and CHD risk is driven by blood pressure.

The study is based on large data sets and succeeds to uncover some novel biology. Generally, the paper is interesting and well-written, the methods seem appropriate and the conclusions are mostly supported by the data. However, some interpretations can be questioned and in some points the presentation could be clearer and more balanced. My comments are as follows:

1) The effect of glycine on CHD risk looks quite modest (OR =0.95 per one SD) and after adjusting for BP it disappears totally. I understand that the authors have focused on discovering novel biology here and not aimed at CHD risk prediction but nevertheless a brief comment on clinical significance of these findings would be appropriate.

We thank the reviewer for this helpful suggestion and have now included a section in the discussion where we discuss potential clinical implications of our findings (see Discussion lines 398-416). We would like to emphasise that our analyses do not give a conclusive indication of the expected effect size of glycine supplementation on cardio-metabolic disease risk, as the genetically predicted odds ratios range from 0.95 to 0.80 (Figure 5), depending on which genetic score was used and (2) more generally, because genetically predicted effect sizes do not necessarily reflect the effects that can be obtained through intervention. It is furthermore of importance to assess the potential risks associated with glycine supplementation prior to setting up trials, as there is indirect evidence that glycine (and/or serine) may promote oncogenesis.

2) The authors seem to have ignored totally the association of glycine with incident diabetes. A recent report on Framingham study participants with normal fasting glucose at baseline described an inverse association of plasma glycine with the risk of incident diabetes. The Framingham authors also carried out a MR analysis and argue for a causal relationship between glycine levels and the risk of incident DM. (Merino J et al. *Diabetologia*. 2018 Jun;61(6):1315-1324). Clearly, the authors should consider DM in the present analyses and comment on the paper of Merino et al.

We thank the reviewer for their suggestion and have now extended the scope of our work to type 2 diabetes (T2D) (results lines 267-315, discussion lines 357-397). Our findings indicate a strong inverse genetic effect of insulin resistance on glycine levels, which may drive the consistently observed association of low glycine levels with incidence of T2D. While we do identify significant effects of specific genetic variants in genes related to glycine catabolism on T2D risk, genetically predicted glycine based on the 24 or 6 SNP score was not associated with CHD. We therefore remain cautious in our interpretation of an overall causal role of glycine levels to diabetes.

Our findings do not entirely replicate the recently reported protective effect of glycine on T2D based on an MR analysis using 5 loci for glycine. The significant inverse genetic association of glycine with T2D risk reported by Merino *et al.* was largely driven by *CPS1*, of which the glycine-raising allele was nominally associated with lower T2D risk in the 11,600 T2D cases and 33,000 controls. Our analyses based on 74,124 T2D cases and 824,006 controls did not replicate the

nominal association with T2D for *CPS1*, nor for the genetic scores that included *CPS1* (discussion lines 379-385).

3) If the effect of glycine on CHD risk is mediated through blood pressure as the authors argue, it is a bit strange that no association was observed between glycine and the risk of stroke. Usually, blood pressure is a stronger risk factor for stroke than for CHD and one would expect also here that the life-long lower BP level in persons with high glycine would be reflected as lower risk of stroke.

We thank the reviewer for raising this important point. If glycine exerted its cardio-protective effect entirely through blood pressure, then a protective effect on stroke, and in particular on haemorrhagic stroke, for which blood pressure is a stronger risk factor than for CHD (Rapsomaniki *et al.*, *The Lancet* 2014, Vol 383, pages 1899-911), would indeed be expected. Given the lack of an association between glycine and stroke incidence and the multiple other mechanisms through which glycine may affect cardio-metabolic disease risk, we do expect that there are other physiological mechanisms through which glycine may influence risk of CHD. We have now acknowledged this more extensively in the discussion and have given a more nuanced interpretation of our findings related to blood pressure (lines 345-350).

For completeness, we have expanded the MR analyses to stroke and stroke sub-types, based on summary-level GWAS results from the MEGASTROKE consortium (Malik *et al.* *Nature Genetics* 2018, Vol. 50, pages 524-537) and genetic associations in UK Biobank. We find no genetic associations between glycine levels and risk of stroke sub-types (lines 207-209, Supplementary Table 7, Supplementary Figure 5).

4) Glycine was measured in different laboratories with different methods and different fasting times: Metabolon mass-spec from non-fasted plasma samples in the EPIC-Norfolk study; AbsoluteIDQ p180 Kit of Biocrates (i.e. mass spec) from fasted plasma samples of the Fenland study; and NMR metabolomics from non-fasting serum samples of the INTERVAL trial. Were any overlapping measurements performed? Do we know anything about the agreement of glycine measurements in different conditions, i.e., plasma vs. serum, fasting vs. non-fasting, and the two different mass-spec platforms vs. the NMR platform? There is no common standardization in the metabolomics field and this raises a concern that the results can vary considerably depending on platform and other measurement conditions.

To assess between-study differences of the genetic loci identified for glycine, we have now tested for heterogeneity based on the Cochran's Q statistic in both the Z score-based meta-analysis of the 5 studies (Supplementary Table 2 column J) and the effect size-based meta-

analysis of the 3 studies for which the effect sizes are on the same scale (Fenland, EPIC-Norfolk and INTERVAL) (Supplementary Table 2 column M). Despite the fact that the metabolomics platforms, sample types and fasting times differed between the studies, no evidence for heterogeneity in the genetic associations between the studies was found, except for the locus in *CPS1* (rs715). This suggests that the genetic determinants of glycine may be largely independent on fasting status, adopted measurement technique and sample type. Because the *CPS1* locus has an unusually high significance ($p=3 \times 10^{-1632}$) and very strong effect size (per-allele beta on standard deviations of glycine=0.444), we think that the identified heterogeneity for the *CPS1* locus may have reached significance despite relatively small differences in effect sizes between the 3 studies for which the effect sizes were on the same scale (Supplementary Figure 1).

We have added Supplementary Table 1 which gives a full overview by study of the metabolomics platforms, sample types, fasting status, etc. and show in Supplementary Figure 1 for each locus the effect sizes by study, for the 3 studies for which the effect sizes are on the same scale and can therefore be directly compared (Fenland/Biocrates, EPIC-Norfolk/Metabolon and INTERVAL/NMR).

For a subset of the INTERVAL participants (N=7,892), metabolite levels were measured on both the Metabolon and NMR platforms. Glycine levels based on the Metabolon and NMR platforms correlated well ($R=0.62$, based on untransformed data). We unfortunately have no measurements of the same samples on the Metabolon and Biocrates platforms or on the NMR and Biocrates platform. However, a study by Yet *et al.* (PLoS ONE, 2016, Vol. 11, No. 4 e0153672) showed that measured glycine levels from 1,000 participants based on the Biocrates and Metabolon platform correlated well ($R=0.71$), and that of the 43 metabolites for which measures were compared between the platforms, glycine measures showed the 7th strongest correlation between platforms.

5) The largest material of the study comes from the UK Biobank, which has a participation rate of about 6% and the participants are known to be healthier and socioeconomically better off than the average UK population. The second largest material, the INTERVAL trial consists of 50,000 blood donors, who also are likely to be healthier than the average population. The authors should at least comment on whether the “healthy participant effect” has any influence on these results.

Thank you for raising this important consideration. We agree that healthy participant bias is a relevant concern for behavioural and observational research but has more recently been hypothesised to also have a potential influence on genetic associations (Munafò *et al.* 2018, International Journal of Epidemiology 47:1, 226-235). We have added a section to the discussion that now specifically addresses this point and the potential implications in our study

(lines 433-442). As Munafò *et al.* suggested, selection bias could theoretically lead to a false positive genetic association between an exposure and an outcome if both the exposure and the outcome influence the likelihood of an individual participating in the study. As individuals are not aware of their glycine levels, any such selection would have to be indirect through other factors that affect glycine levels and bias participation in each of the independent studies included (and participating men and women) in the same direction. We therefore consider the extent to which collider bias is influencing the results of the MR analyses due to a healthy participant to be small, but the possibility is now discussed as a theoretical limitation.

Minor comments:

1) If glycine is causally related to CHD risk, it would be of interest to also analyze the “environmental” determinants of glycine levels and show for comparison the proportion of variance they explain. This should be possible in at least some of the large data sets the authors have.

We assessed the proportion of variance in glycine levels explained by sex, BMI, age, smoking, alcohol consumption and physical activity in 10,475 participants of the Fenland study. Sex was strongly associated with glycine ($p=2.4 \times 10^{-169}$), with men having lower glycine levels than women, and explained 8.6% of the variance in glycine levels. BMI was inversely associated with glycine levels ($p=3.4 \times 10^{-70}$) and explained 3.4% of the variance. Alcohol consumption ($p=4.8 \times 10^{-12}$) and smoking ($p=0.007$) were also inversely associated with glycine levels, but only explained 0.51% and 0.08%, respectively, of the variance in glycine levels. Age ($p=0.79$) and physical activity ($p=0.74$) were not associated with glycine levels (lines 139-142).

2) Fig. 6 and supplementary Figs 1 and 4: It would be clearer to have a vertical line at 0 in these figures.

We have now changed the colour and thickness of the vertical lines through 0 on these 3 figures to make them clearer.

3) How did you take antihypertensive medications into account in the analyses on BP?

For UK Biobank participants who were on anti-hypertensive medication, 15 mmHg and 10 mmHg was added to measured SBP and DBP, respectively. This strategy has been commonly adopted in genetic research on blood pressure traits and follows the recommendations by Tobin *et al.* (Statistics in Medicine, 2005, Vol. 24, Issue 19, Pages 2,911-2,935). We have now added a more complete description of the methods that were used to conduct the GWAS of blood pressure traits in UK Biobank (lines 638-646).

4) P. 22, Cox PH reg modelling: Please describe the models more clearly. Did you exclude

**persons with a history of CHD or stroke at baseline, or are they included in these models?
Were age and sex the only covariates? What about DM, smoking, lipids?**

We have now extended the methods section related to the Cox proportional hazards modelling (lines 681-696). The observational analyses were initially indeed only adjusted for age and sex and prevalent cases were excluded, as our primary aim of this was to compare the observational with the genetic estimate. We have now also fitted fully adjusted models in which a series of cardio-metabolic risk factors were included as covariates (results for CHD/MI: lines 216-223, results for T2D: lines 280-282).

Reviewer #1 (Remarks to the Author):

Assessing the causal association of glycine with risk of coronary heart disease (revised)

Wittemans et al.

This reviewer appreciates the authors comprehensively addressing the points raised in the initial review of the manuscript. There are only a few minor questions that I believe can easily be answered by the authors and which may further improve the manuscript.

1. The authors raise quite an important point about the potentially negative consequences of glycine supplementation in humans as a treatment strategy due to concerns regarding oncogenesis, particularly with respect to colon cancer and lymphoma. Using either publicly available summary level data or any datasets that the authors have access to, it would be of interest to determine whether the same genetic instruments for glycine levels that demonstrate atheroprotective associations yield evidence for a causal association with increase cancer risk.
2. The authors may wish to expand the discussion to point out that glycine per se may not be the actual causal molecule, since various glycine-conjugated molecules, such as N-acetylglycine, propionylglycine, etc, are also strongly associated with SNPs in question.
3. Please provide p-values for men and women in the sex-stratified association with CPS1 shown in Suppl Figure 6.

Reviewer #2 (Remarks to the Author):

The authors have carefully responded to my comments and I have no new comments to add.

Reviewer #1 (Remarks to the Author):

This reviewer appreciates the authors comprehensively addressing the points raised in the initial review of the manuscript. There are only a few minor questions that I believe can easily be answered by the authors and which may further improve the manuscript.

1. The authors raise quite an important point about the potentially negative consequences of glycine supplementation in humans as a treatment strategy due to concerns regarding oncogenesis, particularly with respect to colon cancer and lymphoma. Using either publicly available summary level data or any datasets that the authors have access to, it would be of interest to determine whether the same genetic instruments for glycine levels that demonstrate atheroprotective associations yield evidence for a causal association with increase cancer risk.

We thank the reviewer for their suggestion to expand the scope of our work to cancer and have addressed this point to the extent this was possible given the data available to us. Using summary-level GWAS data made publicly available by the BCAC, OCAC and PRACTICAL consortia, we found no evidence that genetically predicted glycine levels are associated with increased risk of breast, ovarian or prostate cancer. However, consortia data for other cancer sites and types are required to provide genetic evidence against an increased risk for site-specific cancers other than breast, ovarian and prostate, and a thorough assessment of the potential carcinogenicity of glycine is required before glycine supplementation can be considered even in an evaluative setting in human participants. We have added the results of these analyses to the manuscript (Results: lines 526-531 and Supplementary Table 8, Discussion: lines 651-658, Methods: lines 963-973).

2. The authors may wish to expand the discussion to point out that glycine per se may not be the actual causal molecule, since various glycine-conjugated molecules, such as N-acetylglycine, propionylglycine, etc, are also strongly associated with SNPs in question.

As suggested by the reviewer, we have now expanded the section in the discussion related to the reviewer's comment, to emphasise that the observed genetic association of glycine with cardio-metabolic disease outcomes may theoretically be driven by one or several of the glycine-conjugated metabolites, which are associated with the 2 SNP score for glycine (Discussion: lines 697-699).

3. Please provide p-values for men and women in the sex-stratified association with CPS1 shown in Suppl Figure 6.

We have now added the p-values for the associations of rs715 with systolic and diastolic blood pressure in men and women in Supplementary Figure 6.

Reviewer #2 (Remarks to the Author):

The authors have carefully responded to my comments and I have no new comments to add.

We thank Reviewer 2 for their time and for re-evaluating our manuscript, and are glad to read they are satisfied with the manuscript.